# Structural insight on the mechanism of an electron-bifurcating [FeFe] hydrogenase

Chris Furlan[1†], Nipa Chongdar[2†‡], Pooja Gupta[1], Wolfgang Lubitz[2], Hideaki Ogata[3,4], James N Blaza[1*], James A Birrell[2*§]

[1]Structural Biology Laboratory and York Biomedical Research Institute, Department of Chemistry, The University of York, York, United Kingdom; [2]Max Planck Institute for Chemical Energy Conversion, Muelheim an der Ruhr, Germany; [3]Division of Materials Science, Nara Institute of Science and Technology, Ikoma, Japan; [4]Graduate School of Life Science, University of Hyogo, Hyogo, Japan

**\*For correspondence:**
jamie.blaza@york.ac.uk (JNB);
James.Birrell@cec.mpg.de (JAB)

[†]These authors contributed equally to this work

**Present address:** [‡]CSIR-National Institute of Oceanography, Dona Paula, India; [§]University of Essex, Colchester, United Kingdom

**Competing interest:** The authors declare that no competing interests exist.

**Abstract** Electron bifurcation is a fundamental energy conservation mechanism in nature in which two electrons from an intermediate-potential electron donor are split so that one is sent along a high-potential pathway to a high-potential acceptor and the other is sent along a low-potential pathway to a low-potential acceptor. This process allows endergonic reactions to be driven by exergonic ones and is an alternative, less recognized, mechanism of energy coupling to the well-known chemiosmotic principle. The electron-bifurcating [FeFe] hydrogenase from *Thermotoga maritima* (HydABC) requires both NADH and ferredoxin to reduce protons generating hydrogen. The mechanism of electron bifurcation in HydABC remains enigmatic in spite of intense research efforts over the last few years. Structural information may provide the basis for a better understanding of spectroscopic and functional information. Here, we present a 2.3 Å electron cryo-microscopy structure of HydABC. The structure shows a heterododecamer composed of two independent 'halves' each made of two strongly interacting HydABC heterotrimers connected via a [4Fe–4S] cluster. A central electron transfer pathway connects the active sites for NADH oxidation and for proton reduction. We identified two conformations of a flexible iron–sulfur cluster domain: a 'closed bridge' and an 'open bridge' conformation, where a $Zn^{2+}$ site may act as a 'hinge' allowing domain movement. Based on these structural revelations, we propose a possible mechanism of electron bifurcation in HydABC where the flavin mononucleotide serves a dual role as both the electron bifurcation center and as the $NAD^+$ reduction/NADH oxidation site.

## Editor's evaluation

This paper describes a high resolution cryo-EM structure of an [FeFe] hydrogenase purported to operate via an electron bifurcating mechanism. The study aims to resolve a controversy regarding the site of bifurcation through structural characterization of the enzyme complex. The authors propose a mechanism for electron transfer in which conformational changes and cofactor binding events modulate the properties of the pathway.

## Introduction

Electron bifurcation (*Wise et al., 2021*) represents an alternative energy coupling mechanism to the well-known chemiosmotic coupling principle (*Rich, 2003*). Electron bifurcation drives thermodynamically unfavorable (endergonic) redox reactions by coupling them to energetically favorable (exergonic) redox reactions directly within the same enzyme. It achieves this by splitting a pair of electrons from a single two-electron donor to two different spatially separated electron acceptors with one being at

a lower redox potential than the donor and the other being at higher redox potential than the donor. Meanwhile, electron confurcation, the opposite of electron bifurcation takes single electrons from both a high- and low-potential donor and channels both toward an intermediate-potential acceptor. Enzymes using electron bifurcation are found in numerous biochemical pathways including respiration, photosynthesis, methanogenesis, and acetogenesis, where they are crucial for driving important chemical transformations (*Peters et al., 2016*; *Müller et al., 2018*; *Garcia Costas et al., 2017*). The process of electron bifurcation represents an exquisite example of how biochemical systems can use thermodynamic driving forces in a flexible and efficient manner and bifurcating enzymes hold potential as 'molecular transformers' in synthetic biology applications.

Electron bifurcation was first described in the Q-cycle of the respiratory complex III where the two electrons originating from the oxidation of ubiquinol are bifurcated via a *high*-potential pathway to cytochrome c, and via a *low*-potential pathway to reduce ubiquinone to ubiquinol (*Darrouzet et al., 2001*; *Peters et al., 2018*). This process has recently been discovered in a number of other enzymes where an exergonic electron transfer process is used to drive an endergonic one (*Peters et al., 2016*; *Müller et al., 2018*; *Garcia Costas et al., 2017*). Many of these enzymes have been proposed to utilize flavin-based electron bifurcation (FBEB), in which a flavin mononucleotide (FMN) or flavin adenine dinucleotide (FAD) cofactor serves as the branching point for electrons. It first accepts a hydride from an *intermediate*-potential redox couple (typically NAD(P)H) and then sends one electron down a *high*-potential pathway, generating an unstable, low-potential semi-reduced flavin, with strong enough reducing power to send the second electron down a *low*-potential pathway. The importance of FBEB in microbial metabolism and energy conservation is well acknowledged, but its mechanism is still poorly understood, with only a few examples so far being studied in detail, such as butyryl-CoA dehydrogenase-electron-transferring flavoprotein complex (Bcd-EtfAB) and Fd-dependent transhydrogenase (NfnI) (*Buckel and Thauer, 2018a*).

*Thermotoga maritima* is a hyperthermophilic anaerobic eubacterium that is interesting for biohydrogen production due to its ability to produce high levels of hydrogen from a wide range of carbohydrates at elevated temperatures (*Chou et al., 2008*; *Boileau et al., 2016*). The heterotrimeric [FeFe] hydrogenase, HydABC, from *T. maritima* is a soluble cytoplasmic enzyme involved in fermentation. It uses electrons from the one-electron carrier ferredoxin ($E°' \approx -450$ mV *Schut and Adams, 2009*), which is reduced during pyruvate metabolism, and the two-electron carrier NADH ($E°' \approx -20$ mV; *Schut and Adams, 2009*), produced during glucose metabolism, to reduce protons to hydrogen ($E°' \approx -420$ mV; *Schut and Adams, 2009*). The mechanism by which this enzyme functions is debated, however, the predominant view is that an FBEB mechanism is operative (*Buckel and Thauer, 2018b*).

Initially, the site of bifurcation was speculated to be a second flavin cofactor (*Buckel and Thauer, 2013*). However, biochemical studies do not corroborate the presence of a second flavin (*Chongdar et al., 2020*). In another hypothesis, the hydrogen conversion center, the so-called H-cluster, which also undergoes two-electron redox chemistry, was speculated to be the electron bifurcation center (*Peters et al., 2018*). However, spectroscopic studies suggest that the H-cluster of HydABC has redox properties similar to the non-bifurcating [FeFe] hydrogenases, having a stable one-electron reduced state, and is, therefore, also unlikely to be the site of bifurcation (*Chongdar et al., 2020*). This leaves the biochemically characterized FMN at the NADH-binding site as the most likely electron bifurcation center. However, it is unclear how this site can serve as both a two-electron donor to the *high*-potential $NAD^+/NADH$ couple and as a two-electron-bifurcating site from an *intermediate*-potential couple to *high*- and *low*-potential couples.

As structural data would reveal the complex arrangement of redox cofactors in this enzyme and provide a stronger basis for understanding the mechanism of electron bifurcation, here we report a 2.3-Å resolution structure of HydABC based on electron cryo-microscopy (cryo-EM) of single particles. The cryo-EM structure suggests a synergic coupling between two HydABC heterotrimers connected through the His-ligated [4Fe–4S] cluster in the HydA subunit, which may allow functionally important electron exchange between the two heterotrimers. The structure also reveals flexible C-terminal (CT) domains in HydA and HydB (here named 'bridge' domains), which contain additional iron–sulfur clusters. These domains interact through non-covalent interactions and may provide a second electron transfer pathway. Thus, this structure provides details of the arrangement of the redox clusters in HydABC, based on which a novel mechanism of electron bifurcation is proposed in which the FMN in HydB serves two roles: as an $NAD^+$ reduction site and as an electron bifurcation site. We also compare

our results to a recently published structure of a related [NiFe] hydrogenase with a similar arrangement of cofactors around the NADH-binding site (*Feng et al., 2022*).

## Results

### The structure of HydABC

The heterologous production of apo-HydABC in *Escherichia coli* was described recently (*Chongdar et al., 2020*). In our previous work, it was shown that this preparation contains all the redox cofactors of the native HydABC enzyme except for the [2Fe] subcluster of the hydrogenase active site (H-cluster), which *E. coli* is unable to synthesize. In particular, Fe quantitation measurements of the heterologously produced enzyme agreed with the expected number of iron–sulfur clusters based on sequence analysis, and were even higher than those from the native enzyme (*Verhagen et al., 1999*). Furthermore, electron paramagnetic resonance (EPR) spectra of the reduced apo- and reduced holo-HydABC (where the H-cluster is EPR-silent) were identical to each other and the same as those from the native enzyme (*Figure 1—figure supplement 1* and *Verhagen et al., 1999*). A drawback of using this apo-HydABC preparation is that we cannot observe how the structure is affected by reduction by $H_2$.

Here, we have used this heterologously expressed apo-HydABC to prepare the cryo-EM grids under air, as apo-HydABC lacking the [2Fe] subcluster is much less oxygen sensitive. Previous studies have indicated that the incorporation of the [2Fe] subcluster minimally affects the structure of [FeFe] hydrogenases (*Esselborn et al., 2016*) (except for the enzyme from *Chlamydomonas reinhardtii*; *Mulder et al., 2010*) and, as shown by our structure, the H-cluster is located far away from the likely electron bifurcation site. Following grid imaging, data collection (*Figure 1—figure supplement 2*), and processing (*Figure 1—figure supplement 3*), we obtained a 2.3-Å resolution map when D2 symmetry was enforced (*Figure 1*, *Video 1*, *Figure 1—figure supplement 4*, *Figure 1—figure supplement 5*). Into this, an atomic model of HydABC was constructed, starting with a homology model based on homologous subunits in bacterial complex I (*Chongdar et al., 2020*; *Baradaran et al., 2013*), together with ab initio model building in regions of the highest resolution (*Figure 1—figure supplement 6*). Initially, the last 91 and 61 CT residues of HydA and HydB, respectively, could not be built as they were not present in the homology model (because complex I does not contain homologous domains) and had a low resolution in the map, indicating regions of high heterogeneity (explored later).

The processed cryo-EM map shows that HydABC forms a dodecameric complex, $Hyd(ABC)_4$, composed of a tetramer of HydABC heterotrimer units (from now on referred to as protomers). Oligomerization of HydABC occurs through interactions between four HydA subunits in the core of the complex (*Figure 1A* and *Video 1*). Each HydA has extensive interactions with one adjacent HydA chain (buried surface area of 2280 Å$^2$), and minor interactions with another HydA chain (780 Å$^2$) (*Figure 1A*). HydB is tightly bound to a single HydA (buried surface area of 1232 Å$^2$, *Supplementary file 1*) but with minor interactions between HydB of one heterotrimer and HydA and HydB in another heterotrimer. HydB and HydC extend outward from the core and form the four lobes clearly visible in the 2D class averages (*Figure 1—figure supplement 2*). The HydA core is the best resolved part of the map, consistent with the core being rigid and homogenous (*Figure 1—figure supplement 5*).

Based on the density map, each HydABC protomer appears to contain nine redox cofactors including five [4Fe–4S] clusters (one of which forms the [4Fe–4S] subcluster of the H-cluster), three [2Fe–2S] clusters, and one FMN. However, based on published Fe quantitation as well as published sequence analysis predictions we expect a total of seven [4Fe–4S] clusters (including the subcluster of the H-cluster) and four [2Fe–2S] clusters in each HydABC protomer (*Verhagen et al., 1999*; *Verhagen and Adams, 2001*). According to sequence predictions, these missing clusters should be located in the less well-resolved CT regions of the HydA and HydB subunits (discussed below) (*Verhagen et al., 1999*). Interestingly, a high-density site, likely a monometallic center, is found in the resolvable part of the HydB-CT domain, at the end of a small four-helix bundle. Inductively coupled plasma mass spectrometry on the separately produced and purified HydB subunit identified 0.99 ± 0.43 Zn/protein and ≈14.2 ± 1.5 Fe/protein. As the observed Fe content matches with the estimated Fe content of HydB, which is expected to contain three [4Fe–4S] clusters and one [2Fe–2S] cluster (14 Fe/protein), these results allow us to assign the metal center as zinc ($Zn^{2+}$). This is further supported by the identities

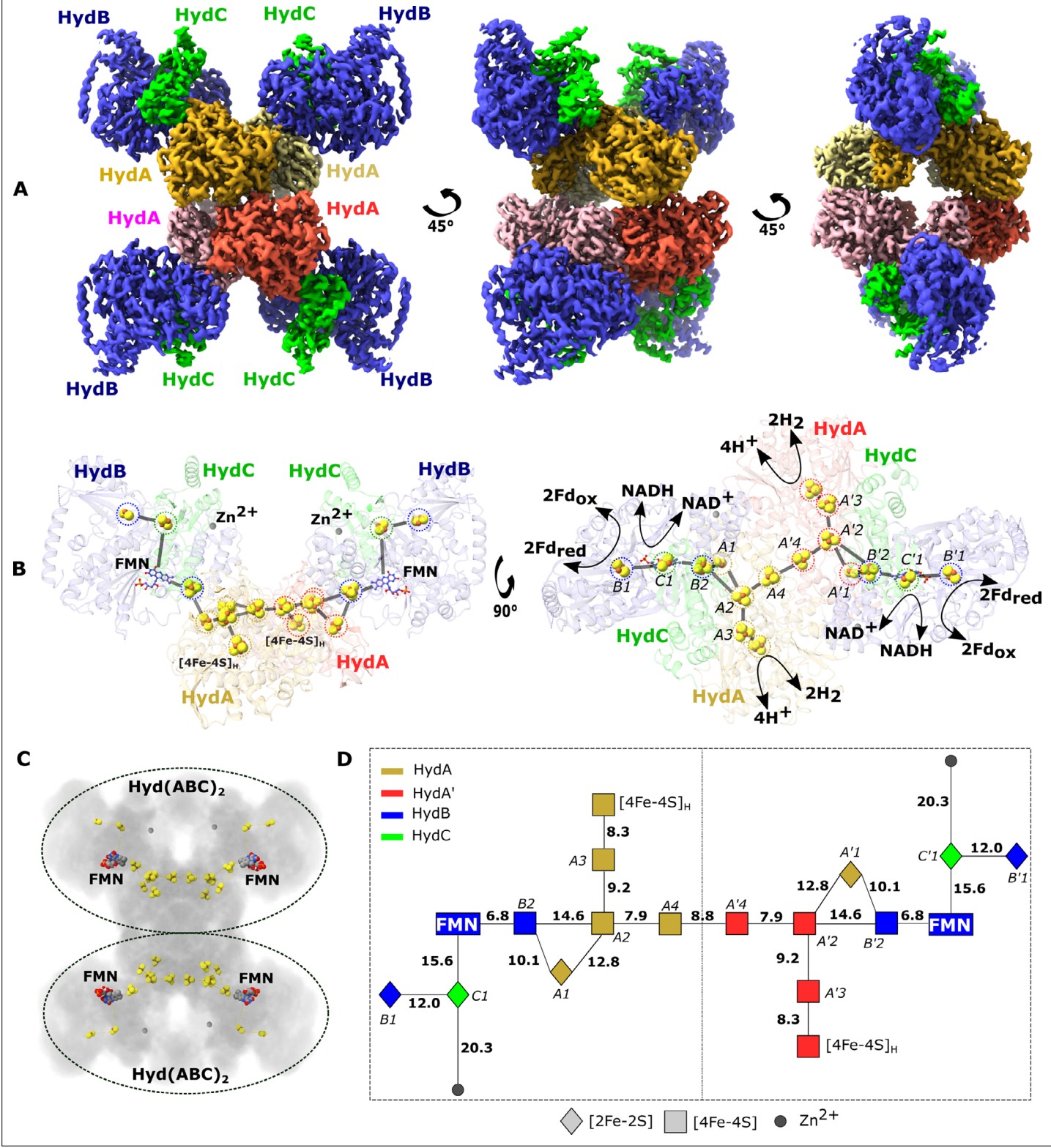

**Figure 1.** Cryo-EM structure of the HydABC tetramer and arrangement of the redox cofactors. (**A**) The unsharpened 2.3 Å map of Hyd(ABC)$_4$ with D2 symmetry enforced showing a tetramer of HydABC heterotrimers. All four copies of HydB and C are colored blue and green, respectively. The four HydA copies that make up the core of the complex are in orange, yellow, pink, and red. The top and bottom halves of the complex are constituted by dimers of HydABC protomers (each HydABC unit is a protomer); the two protomers within the same dimer are strongly interacting, while a weaker interaction is present between the top and bottom dimers. (**B**) HydABC dimer highlighting the iron–sulfur clusters and flavin mononucleotide (FMN)

*Figure 1 continued on next page*

*Figure 1 continued*

constituting the electron transfer network. (**C**) The arrangement of redox cofactors within the protein complex, showing two independent identical redox networks (dashed circles); each redox network is composed of iron–sulfur clusters belonging to a Hyd(ABC)$_2$ unit composed of two strongly interacting HydABC protomers. (**D**) Schematic of the electron transfer network of one of the two identical Hyd(ABC)$_2$ units showing edge-to-edge distances (in Å) between the various cofactors. Note that our structure is of apo-HydABC and contains only the [4Fe–4S]$_H$ subcluster of the H-cluster. The 2H$^+$/H$_2$ interconversion reaction in (**B**) illustrates the site at which this reaction occurs, but this will only occur in the full assembled H-cluster including [2Fe]$_H$.

The online version of this article includes the following figure supplement(s) for figure 1:

**Figure supplement 1.** X-band (CW) electron paramagnetic resonance (EPR) spectroscopy on the isolated HydB subunit from *Thermotoga maritima* produced in *E. coli* (0.2 mM) reduced with 10 mM sodium dithionite.

**Figure supplement 2.** Micrograph and 2D classes averages of *T. maritima* HydABC.

**Figure supplement 3.** Classification and refinement of the cryo-EM density map for *Tm*HydABC using the RELION pipeline.

**Figure supplement 4.** Fourier shell correlation (FSC) curve showing the resolution of the D2 map, reaching 2.3 Å.

**Figure supplement 5.** Local resolution of the D2 map.

**Figure supplement 6.** Exemplar denisty maps and models for various regions of the TmHydABC cryo-EM structure.

**Figure supplement 7.** Investigating alternative models for the observed map density (average resolution of 3 Å) for the metal center in the HydABCSL from *Acetomicrobium mobile*.

---

of the ligating residues: three cysteines and one histidine in a tetrahedral coordination geometry (*Figure 1—figure supplement 6*; *Ireland and Martin, 2019*).

In a related electron-bifurcating [NiFe] hydrogenase (HydABCSL) from *Acetomicrobium mobile* this monometallic site was modeled as a [2Fe–2S] cluster with five coordinating ligands from the protein (*Feng et al., 2022*). Furthermore, an oxygen-tolerant [FeFe] hydrogenase (*Cb*A5H) from *Clostridium beijerinckii* contains a similar domain and was suggested to ligate a [4Fe–4S] cluster (*Winkler et al., 2021*). We have compared our structure with these two previously published structures and find it is possible to replace the FeS clusters with a Zn (or other similarly sized tetrahedral metal center) and obtain a reasonable model; given the ~3 Å resolution it is not possible to confidently distinguish which fits better. *Figure 1—figure supplement 7* shows the details of one such model for HydABCSL. EPR spectra of the separately produced HydB subunit (*Figure 1—figure supplement 1*) are identical to those published for the HydB subunit obtained from the native *T. maritima* (*Verhagen et al., 1999*), confirming that the native and heterologously produced HydB subunits contain the same cohort of EPR active FeS clusters. Furthermore, our HydABC preparation is fully active in electron bifurcation (*Chongdar et al., 2020*). These results indicate that *Tm*HydABC contains a single metal at this site and not a [2Fe–2S] cluster. Regardless, it would appear that the cofactor bound at this site does not transfer electrons in *Tm*HydABC.

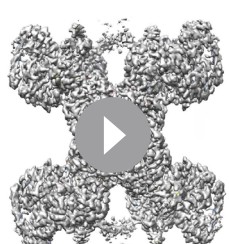

**Video 1.** In the first few frames the cryo-EM structure of the heterododecameric Hyd(ABC)$_4$ complex overlaid (7P5H) with the cryo-EM map can be seen rotating around the central vertical axis. The cryo-EM map then fades to reveal the structural model with the protein represented in the ribbon mode (HydA in the center in red, pink, green, and orange, HydB in blue, and HydC in yellow) and the cofactors shown as spheres. After rotation, again around the central vertical axis, the ribbon structure fades to reveal the iron–sulfur cluster cofactors as yellow and brown spheres, the zinc sites as gray spheres, and the flavin mononucleotide (FMN) as sticks. It is clear to see that the redox cofactors in the top and bottom halves of the dodecamer are separated by a large distance, too large for efficient electron transfer. Thus, the two redox cofactor networks cannot exchange electrons with each other. It is also clear that there is a core electron transfer pathway linking the FMN sites and a peripheral electron transfer pathway consisting of two iron–sulfur clusters on the other side of the FMN from the core pathway. Finally, the video centers on the region around one of the FMN cofactors for a closer view.

https://elifesciences.org/articles/79361/figures#video1

## Cofactor arrangement in HydABC

Electron transfer chains, often connecting distant active sites, are composed of redox-active cofactors usually less than 14 Å apart to allow sufficiently fast electron tunneling through the protein dielectric to sustain physiological processes

(*Page et al., 1999*). In each HydABC heterotrimer, the spatially distant H-clusters and FMN centers are connected via a chain of four FeS clusters (A1, A2, A3, and B2, see *Figure 1D* for cluster nomenclature). The edge-to-edge distances between all these clusters are <15 Å and within a distance for electron transfer at physiologically relevant rates (*Figure 1D*). Among the three remaining FeS clusters, the [4Fe–4S] cluster from HydA (A4) lies at the interface of the two tightly interacting HydA chains, and the two [2Fe–2S] clusters from HydC (C1) and HydB (B1) subunits lie in the vicinity, but on the opposite side, of the FMN.

Within the Hyd(ABC)$_4$ complex, there appear to be two redox networks, each composed of two HydABC protomers with an extended electron transfer chain, separated by at least 50 Å and held together by extensive HydA–HydA interactions (*Figure 1B, C*). The large distance between each electron-transfer network indicates there is no possibility for electrons to be exchanged and that they probably function independently (*Figure 1C*). The two tightly interacting HydABC protomers within the Hyd(ABC)$_2$ unit are connected (8.8 Å separation) through the His-ligated [4Fe–4S] cluster (A4) in HydA (*Figure 1B*), part of the so-called Y-junction of iron–sulfur clusters (*Zuchan et al., 2021*). This junction is well conserved in a wide number of structurally related enzymes, but its significance is unknown. In HydABC it is clear that the Y-junction connects the NADH and ferredoxin oxidation sites to the hydrogenase active site and to the neighboring protomer. The two A4 clusters are separated by 9.0 Å and have the possibility to allow an overflow of electrons from one protomer to the other. An electronic connection between two identical protomers has already been observed in cytochrome $bc_1$ (*Swierczek et al., 2010*), called an electronic 'bus-bar', which is speculated to have a number of possible roles such as allowing the physiological function of the protein even after operational damage of one of the two protomers. This connection does not provide a 'short circuit', however, since in HydABC the A4 clusters simply connect FMN and H-cluster sites from adjacent protomers that are already directly connected within their respective protomer.

## Structural comparison of HydABC with homologous proteins

The spatial arrangement of subunits HydA, B, and C in the HydABC protomer is similar to that of subunits Nqo3, Nqo1, and Nqo2, respectively, in the NADH oxidation (N) module of *Thermus thermophilus (Tt)* respiratory complex I (*Figure 2—figure supplement 1*). This comparison is useful because complex I is structurally well characterized, but does not oxidize ferredoxin or carry out electron bifurcation. Therefore, structural differences between the subunits of complex I and HydABC may reveal important insight into the mechanism of electron transfer in the latter. The individual subunits are structurally highly similar and here we use RMSD (root-mean-square deviation between the Cα positions in homologous pairs of amino acids) as a quantitative measure of similarity between proteins. The highest similarity is between HydB and Nqo1 (RMSD of 1.040 Å) (*Gutiérrez-Fernández et al., 2020*), followed by HydC and Nqo2 (RMSD 1.152 Å), and the lowest similarity between HydA and Nqo3 (RMSD 1.294 Å) (*Figure 2A*). The remarkable structural similarities between HydB and Nqo1 subunits agree with their common evolutionary origins (*Schut et al., 2013*) and suggest that NADH oxidation follows a similar mechanism in both enzymes (*Figure 2B*). The structural differences between Nqo3 and HydA likely reflect the fact that the latter accommodates the hydrogenase H-cluster and facilitates oligomerization of the Hyd(ABC)$_4$ complex. It should be emphasized here that our structure of HydABC was produced from an enzyme lacking the [2Fe] subcluster component of the H-cluster. However, previous studies have shown negligible structural changes of the protein upon [2Fe] subcluster incorporation (*Esselborn et al., 2016*).

The structural similarities between HydABC and *Tt* respiratory complex I are also reflected by the FeS cluster positioning that is in excellent agreement in these two proteins (*Figure 2C*). However, in contrast to the *Tt* complex I, the HydABC protomers contain five additional FeS clusters. One of these additional clusters is a [4Fe–4S] cluster (A3) that connects the [4Fe–4S] subcluster of the H-cluster (analogous to the cluster N7 in *Tt* complex I) with the rest of the electron transfer network (<10 Å separation from both). Another additional cluster is a [2Fe–2S] cofactor in HydB (B1) that is 12 Å from the [2Fe–2S] cluster in HydC (C1, analogous to N1a in *Tt* complex I); due to this connection and the proximity of HydC to the 'bridge' domains (discussed later) it is likely that the [2Fe–2S] cluster in HydC has an important role in the mechanism of electron bifurcation. This is in contrast to its analogous N1a cluster in complex I, the role of which is unclear but is certainly not part of the main catalytic electron transfer pathway (*Birrell et al., 2013*; *Gnandt et al., 2017*). Lastly, the Zn$^{2+}$ site in HydB is

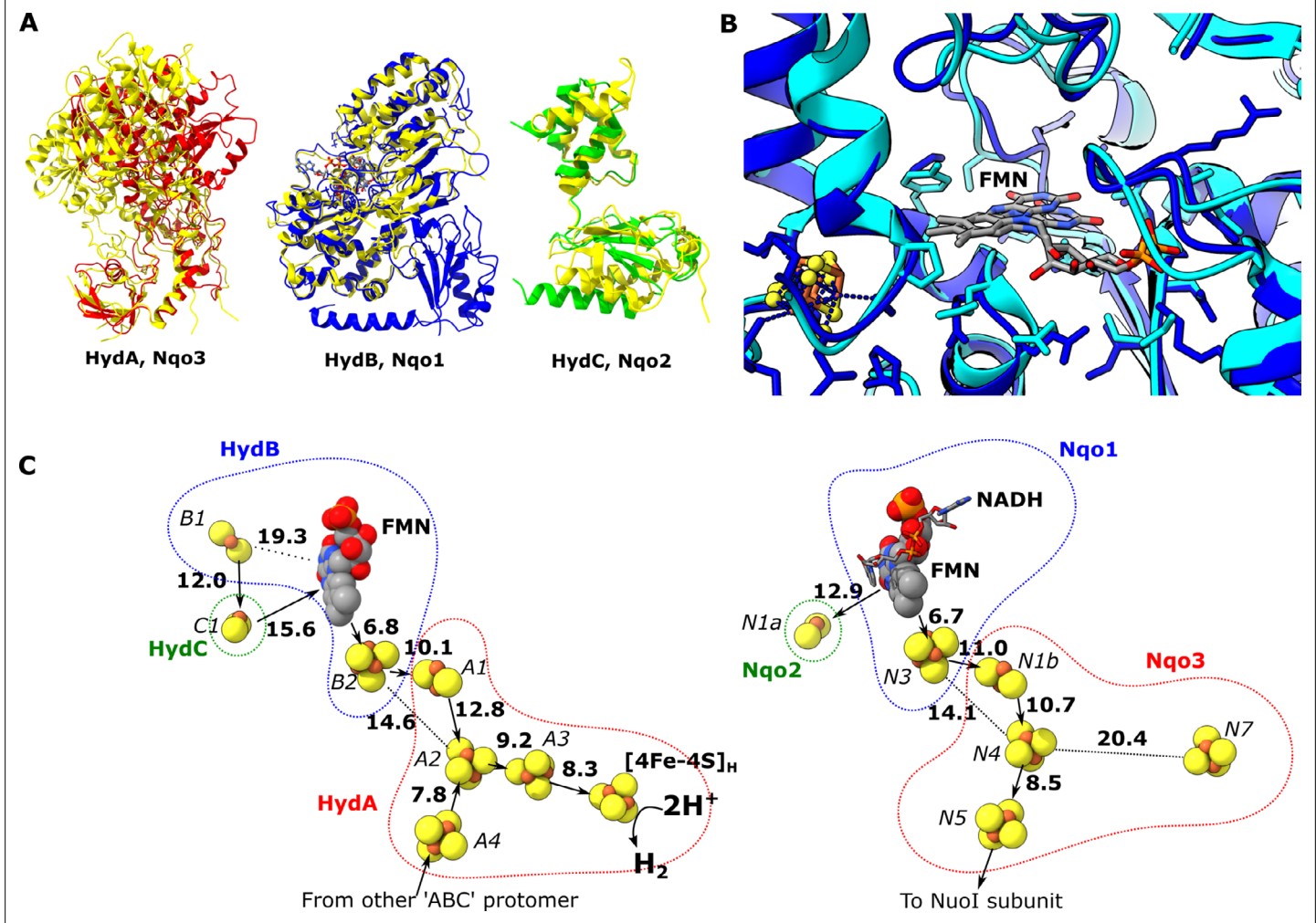

**Figure 2.** Comparion of the HydA, B and C subunits of the electron bifurcating [FeFe] hydrogenase from *Thermotoga maritima* with the Nqo3, 1 and 2 subunits from respiratory complex I from *Thermus thermophilus*. (**A**) Subunits HydA (red), HydB (blues), and HydC (green) overlaid with, respectively, Nqo3, Nqo1, and Nqo2 (all yellow) of complex I from *T. thermophilus* (***Gutiérrez-Fernández et al., 2020***, PDB: 6ZIY). (**B**) Comparison of the NADH-binding site of the Nqo1 subunit of complex I from *T. thermophilus* (light blue) with the flavin mononucleotide (FMN) site in HydB; the high similarity suggests NADH binds in the proximity of FMN in HydABC similar to complex I. (**C**) Electron transfer network in HydABC compared to complex I from *T. thermophilus* with edge-to-edge distances indicated in bold. The red, blue, and green dotted lines indicate the cofactors present in the HydA (Nqo3), HydB (Nqo1), and HydC (Nqo2) subunits, respectively. Note that our structure is of the apo-HydABC and lacks the [2Fe]$_H$ subcluster of the H-cluster. The 2H$^+$/H$_2$ interconversion reaction in (**C**) illustrates the site at which this reaction occurs, but this will only occur in the full assembled H-cluster including [2Fe]$_H$.

The online version of this article includes the following figure supplement(s) for figure 2:

**Figure supplement 1.** HydABC protomer next to Nqo3, Nqo1, and Nqo2 subunits of complex I from *Thermus Thermophilus* (PDB ID: 6I1P) in their native arrangements (***Gutiérrez-Fernández et al., 2020***).

not conserved in Nqo1, instead of three Cys and one His the homologous amino acids in Nqo1 are Ser, Leu, Arg, and Pro.

The HydA subunit has close structural homology (35% sequence identity) to the well-characterized monomeric non-bifurcating [FeFe] hydrogenase from *Clostridium pasteurianum*, *Cp*I. In contrast to electron-bifurcating [FeFe] hydrogenases, non-bifurcating [FeFe] hydrogenases use a single redox partner, typically ferredoxin. Aligning the two enzymes (using holo-*Cp*I containing the [2Fe] subcluster) shows high similarity (rmsd 1.119 Å) and excellent conservation of the FeS clusters, including the A4 cluster, which connects neighboring HydA subunits in HydABC (***Figure 3***). However, in *Cp*I, for which ferredoxin is the only redox partner, the cluster homologous to A4 is thought to lead to the ferredoxin-binding site (***Artz et al., 2017***), although a study on the related enzyme from *Clostridium*

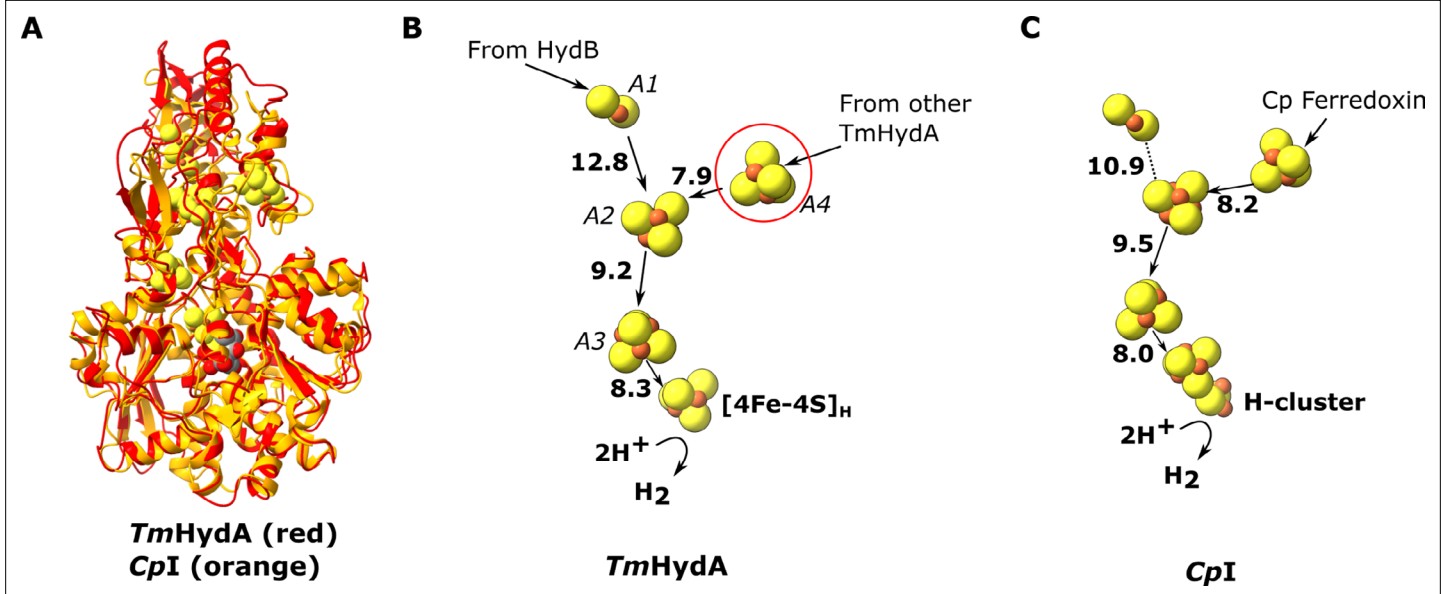

**Figure 3.** Comparion of the HydA subunit of the electron bifurcating [FeFe] hydrogenase from *Thermotoga maritima* with the [FeFe] hydrogenase (CpI) from *Clostridium pasteurianum*. (**A**) HydA from *Thermotoga maritima* (red) compared with *Cp*I hydrogenase from *Clostridium pasteurianum* (orange) (*Artz et al., 2020*, PDB: 6N59). (**B**) Electron transfer network in HydA showing the iron–sulfur cluster that connects adjacent HydABC protomers (red circle). (**C**) Electron transfer network in *Cp*I, with *Cp* ferredoxin, predicted to bind closely to the [4Fe–4S] cluster on the right (*Artz et al., 2017*), although the [2Fe–2S] cluster has also been suggested (*Gauquelin et al., 2018*). Note that only the [4Fe–4S]$_H$ subcluster of the H-cluster is present in our *Tm*HydA structure, whereas the complete H-cluster including the [2Fe]$_H$ subcluster is present in the *Cp*I structure. Edge-to-edge electron transfer distances are indicated in bold. The 2H$^+$/H$_2$ interconversion reaction in (**B**) illustrates the site at which this reaction occurs, but this will only occur in the full assembled H-cluster including [2Fe]$_H$.

The online version of this article includes the following figure supplement(s) for figure 3:

**Figure supplement 1.** Comparison of the region around the H-cluster in the [FeFe] hydrogenase from *Clostridium pasteurianum* (CpI, gray, PDB ID 6N59) with the region around [4Fe–4S]$_H$ in apo-HydABC (pink).

*acetobutylicum* (CaHydA) came to a different conclusion (*Gauquelin et al., 2018*). The multimerization of HydA blocks this site, so the two enzymes must have different ferredoxin-binding sites. This rearrangement is an example of how closely related systems may have different electron transfer pathways formed by different multimerization of their subunits. Importantly, the structure around the H-cluster is highly conserved between CpI and HydABC with only very small deviations in the positions of serveral conserved side chains (*Figure 3—figure supplement 1*).

## A bridging domain formed by the flexible C-termini of the HydA and HydB subunits

The core of the tetrameric HydABC complex is very well resolved, reaching a local resolution of 2.2 Å. However, the lobes formed from HydA and HydB subunits have substantially lower local resolution (~3 Å), due to increased heterogeneity (*Figure 1—figure supplement 5*) and low intensity, blurred map density was observed between the lobes of connected HydABC protomers (*Figure 4—figure supplement 1A*). To investigate the blurred regions, symmetry expansion followed by classification was explored to separate the different conformations into classes. Initial attempts to use D2 symmetry, to match the core, resulted in maps no better than before, however, using C2 symmetry revealed two classes with bridging density between the HydB lobes (*Figure 4A*) with local resolution similar to the lobes formed from HydA and HydB (*Figure 4B*). This bridging density breaks the rotational symmetry between the protomers in the Hyd(ABC)$_2$ unit, explaining why D2 symmetry expansion was ineffective. The two classes correspond to the bridge domain being formed between different HydB lobes: when rotated by 180°, the bridges are identical (*Figure 4A, C*). Despite extensive attempts, we were unable to find a class with both bridges in the closed conformation. The observation that both bridges cannot close simultaneously suggests that these behave as reciprocating elements. A similar observation was made previously for the Rieske domains in the bifurcating $bc_1$ complex (*Maldonado et al., 2021*).

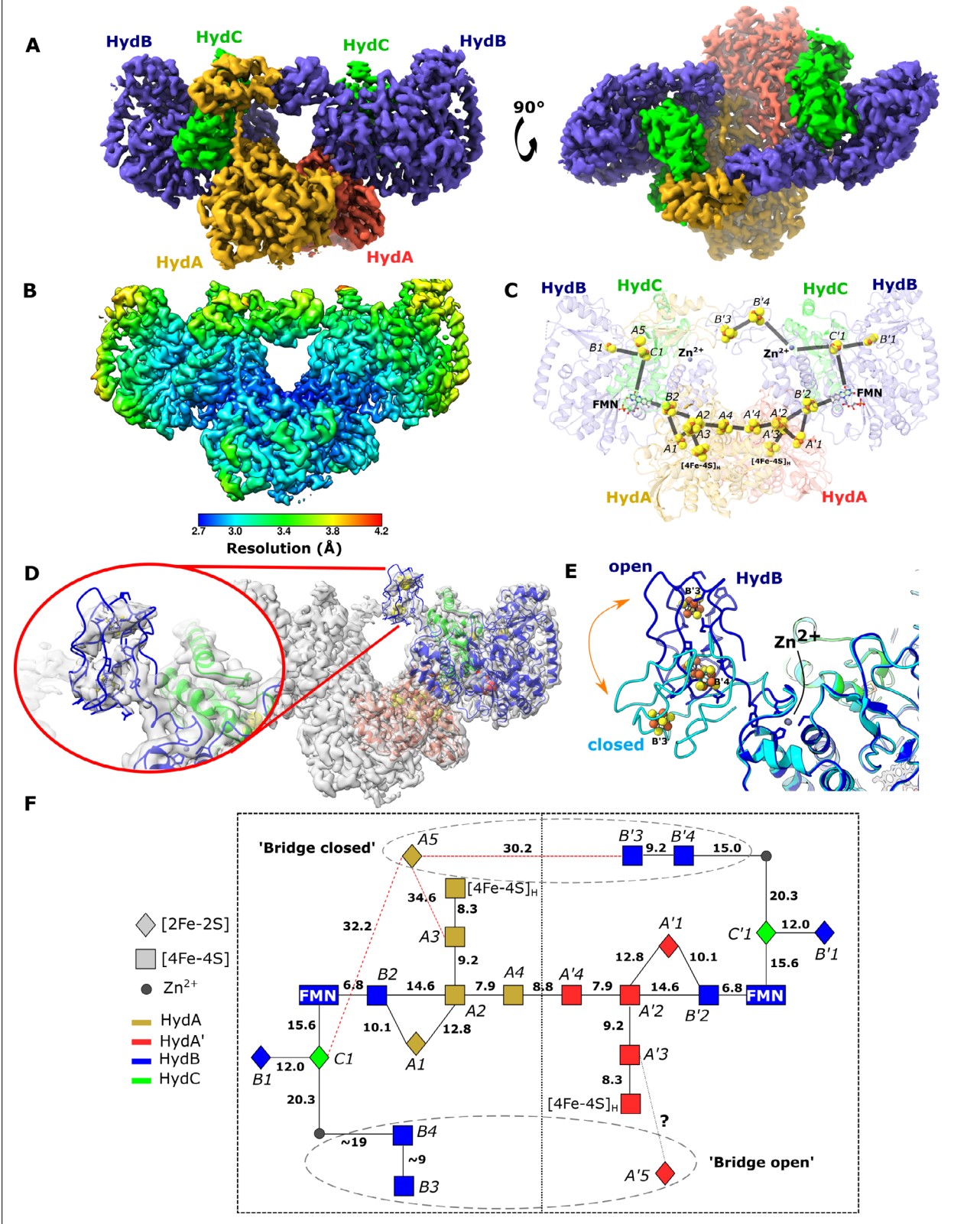

**Figure 4.** Cryo-EM structures of the closed-bridge and open-bridge conformations of HydABC from *Thermotoga maritima*. (**A**) The unsharpened 2.8 Å map of the bridge forward class subparticle, identical to the bridge backward class if a C2 rotation is applied. The map shows only the Hyd(ABC)$_2$ unit as the two Hyd(ABC)$_2$ units constituting the Hyd(ABC)$_4$ complex were found to be independent after 3D classification. All four copies of HydB and C are colored blue and green, respectively. The two HydA copies are in light brown and light red. (**B**) Local resolutions were estimated using the

*Figure 4 continued on next page*

*Figure 4 continued*

local resolution function in RELION with default parameters. (**C**) The atomic model that was built into the map density with the iron–sulfur electron transfer chain. (**D**) Map showing the HydB bridge domain in the open position and its fitted model. (**E**) $Zn^{2+}$ hinge region, showing the two possible conformations of the HydB bridge domain, open (blue) and closed (light blue). (**F**) Schematic of the electron transfer network of one of the two identical $Hyd(ABC)_2$ units showing edge-to-edge distances (Å) between the components. Represented are the iron–sulfur clusters, $[4Fe–4S]_H$ subcluster of the H-cluster, flavin mononucleotide FMN, and $Zn^{2+}$ site; the bridge components and Zn site are enclosed in a dashed ellipse. Each of the two HydABC protomers constituting the $Hyd(ABC)_2$ unit is included within a dashed rectangle. Here, the top bridge is represented in its closed conformation, while the bottom one is in its open conformation. Note that our structure is of the apo-HydABC and lacks the $[2Fe]_H$ subcluster of the H-cluster.

The online version of this article includes the following figure supplement(s) for figure 4:

**Figure supplement 1.** Classification and refinement of the symmetry expanded cryo-EM density maps for *Tm*HydABC using the RELION pipeline.

**Figure supplement 2.** Fourier shell correlation (FSC) curves of the symmetry expanded maps.

To further explore the particles without a bridge a further classification was used (*Figure 4—figure supplement 1B*). It was possible to obtain a low-resolution map of a class where the HydB CT domain was found in an 'open' conformation (*Figure 4D*). The movement of the HydB C-terminal domain between the bridge open and bridge closed classes is shown in *Figure 4E* and *Video 2*.

In the bridge-containing structure, the two C-terminal [4Fe–4S] clusters (named B3 and B4, *Figure 4F*) of HydB are close enough to exchange electrons with each other but are too far from the next nearest FeS clusters, such as cluster C1 (≈35 Å away) or cluster A5 (≈32 Å away). Furthermore, cluster A5 is completely isolated from exchanging electrons with all the nearest clusters being >30 Å away. Thus, unless the HydA and HydB bridge domains undergo substantial conformational changes, the FeS clusters A5, B3, and B4 cannot participate in electronic exchange with the rest of the enzyme.

The bridge structure is particularly interesting as it appears that the C-terminal cysteine residues of HydB responsible for coordinating [4Fe–4S] clusters in the bridge are conserved in all biochemically characterized electron-bifurcating [FeFe] hydrogenases (*Losey et al., 2017*; *Losey et al., 2020*) suggesting that these clusters play an important role in the electron bifurcation mechanism. However, they all lack the analogous part of the bridge domain in HydA, which contains the A5 cluster, which suggests that this cluster may not be a crucial component for electron bifurcation.

## Discussion

In FBEB, two electrons are transferred to the flavin at *intermediate* redox potential in the form of a hydride, and the electrons are split so that one electron goes along a *high*-potential pathway and the other goes along a *low*-potential pathway. HydABC is not a typical flavin-based electron-bifurcating enzyme. The FMN in HydABC exchanges electrons with $NAD^+/NADH$, which forms the *high*-potential couple ($E°' ≈ -320$ mV), and exchanges electrons with the H-cluster, which in turn exchanges electrons with $2H^+/H_2$, the *intermediate*-potential couple ($E°' ≈ -420$ mV), while oxidized/reduced ferredoxin, the *low*-potential couple ($E°' ≈ -450$ mV), appears to exchange electrons with a separate pathway. The hypothesis that a second flavin site is responsible for electron bifurcation (*Buckel and Thauer, 2013*) is neither supported by previous biochemical experiments (*Chongdar et al., 2020*), nor by the cryo-EM structure of HydABC presented herein: only a single flavin (the FMN in HydB) that accepts a hydride from NADH exists in this enzyme. Another hypothesis is that the H-cluster is the bifurcation center (*Peters et al., 2018*). However, the H-cluster of HydABC shows similar redox behavior to the H-cluster from non-bifurcating

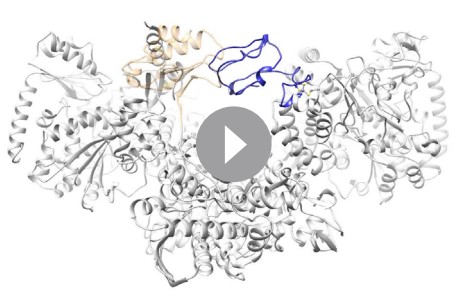

**Video 2.** In this movie, the conformational change observed between the 'Bridge closed forward' (7P8N) and 'Open bridge' (7PN2) classes is shown. The HydB C-terminal iron–sulfur cluster domain is colored blue and the HydA C-terminal iron–sulfur cluster domain is colored orange. The zinc ion (gray sphere) and ligating residues (three cysteine ligands and one histidine) are also shown. The location of the HydA C-terminal domain when the bridge is open is unknown so it is shown transparently in both states for reference.
https://elifesciences.org/articles/79361/figures#video2

[FeFe] hydrogenases (*Chongdar et al., 2020*). In addition, a structural comparison of the HydA subunit (of HydABC) with the non-bifurcating [FeFe] hydrogenase *Cp*I reveals that the primary and secondary coordination spheres of the H-cluster are highly conserved in the two enzymes, thereby, supporting our previous conclusion that the H-cluster is also not the bifurcation center (*Chongdar et al., 2020*). Lastly, the H-cluster is located at the end of an electron transfer pathway rather in the middle of one, which makes it a very unlikely branch site.

By excluding that the H-cluster or a second flavin function as bifurcation sites, and since our new cryo-EM structures reveal that there are no other possible electron bifurcation sites, we are left with the possibility that the FMN in HydB is indeed the electron bifurcation site. Our first structure reveals that the FMN is located at a branch point connecting the core electron transfer pathway from the H-cluster and the additional iron–sulfur clusters B1 and C1, while our additional structures reveal that the FMN is close to a zinc site and a mobile iron–sulfur cluster domain, all indicating that it is ideally located for behaving as an electron bifurcation center. However, the FMN must bifurcate electrons in an unprecedented way, since it must also serve as the two electron donor/acceptor of NAD$^+$/NADH. We propose a potential mechanism of electron transfer in HydABC in which the chemistry of the FMN is dependent on nucleotide binding and conformational changes of the HydB-CT domain. This domain, carrying the B3 and B4 clusters, is found in all characterized electron-bifurcating [FeFe] hydrogenases but is absent in non-bifurcating NAD$^+$-dependent multimeric [FeFe] hydrogenases (*Losey et al., 2017*; *Losey et al., 2020*). Therefore, these clusters are believed to be an essential component of the mechanism. The crucial requirements for any proposed mechanism are the following experimental observations:

1. Thermodynamically favorable H$_2$ production from ferredoxin oxidation is prevented in the absence of NADH oxidation
2. Thermodynamically favorable NAD$^+$ reduction by H$_2$ is prevented in the absence of ferredoxin reduction
3. Thermodynamically favorable ferredoxin oxidation by NAD$^+$ is prevented

Electron transfer pathways can be 'broken' in one of two ways: by spacially separating two electron transfer centers or by separating their potentials. Observation 1 may be achieved by spatially separating the ferredoxin oxidation site from the H-cluster. If the HydB-CT with the B3 and B4 clusters is the site of ferredoxin oxidation then these clusters are already separated from the main electron transfer pathway in all of the structures we have presented here. Thus, ferredoxin oxidation by the B3 and B4 clusters would load electrons into the enzyme, ready for transfer to the H-cluster. However, the FMN, the site of NAD$^+$ reduction, is directly connected to the H-cluster via the core electron transfer pathway. Thus, observation 2 can only be achieved through redox potential differences. One possibility is that a cluster in the electron transfer pathway from the H-cluster to the FMN has a (1) very negative or (2) very positive redox potential, limiting the electron transfer rate. However, it is hard to see how this could be used to permit reduction of ferredoxin while hindering reduction of NAD$^+$. A more likely scenario is that the enzyme takes advantage of the FMN's two electron chemistry. By stabilizing the first one-electron reduction potential, but destabilizing the second one-electron reduction potential, the FMN would effectively become a one-electron transfer center incapable of NAD$^+$ reduction to NADH. This could be regulated by the movement of the HydB-CT domain such that conformational changes upon reduction of ferredoxin would destabilize the one-electron reduced FMN, forcing it to oxidize a nearby cluster and become two-electron reduced and NAD$^+$ reduction competent. Observation 3 would be achieved by a combination of the spatial separation of the ferredoxin oxidation and NAD$^+$ reduction sites, as well as the stabilization of the first one-electron redox potential of the FMN.

A potential mechanism would operate as follows:

During the oxidation of H$_2$ to reduce NAD$^+$ and ferredoxin (electron bifurcation) (*Figure 5*), (1) four electrons from the oxidation of two H$_2$ molecules at the H-cluster travel via the core electron transfer pathway composed of the A1, A2, A3, and B2 clusters toward FMN. At first, the one-electron redox potential for the FMN ($E_{\text{FMN/FMN}^{\bullet-}}$) is too negative for the formation of the FMN$^{\bullet-}$ radical. Since the B2 cluster is at the end of the four-helix bundle connected to the Zn site, reduction of this cluster could trigger the opening of the HydB-CT domain. (2) NAD$^+$ binding to the FMN increases $E_{\text{FMN/FMN}^{\bullet-}}$ allowing the formation of the FMN$^{\bullet-}$ radical, but not full reduction to FMNH$^-$. NAD$^+$ binding also stabilizes a conformation of the HydB-CT 'bridge' domain in which the B3 and B4 clusters are close to

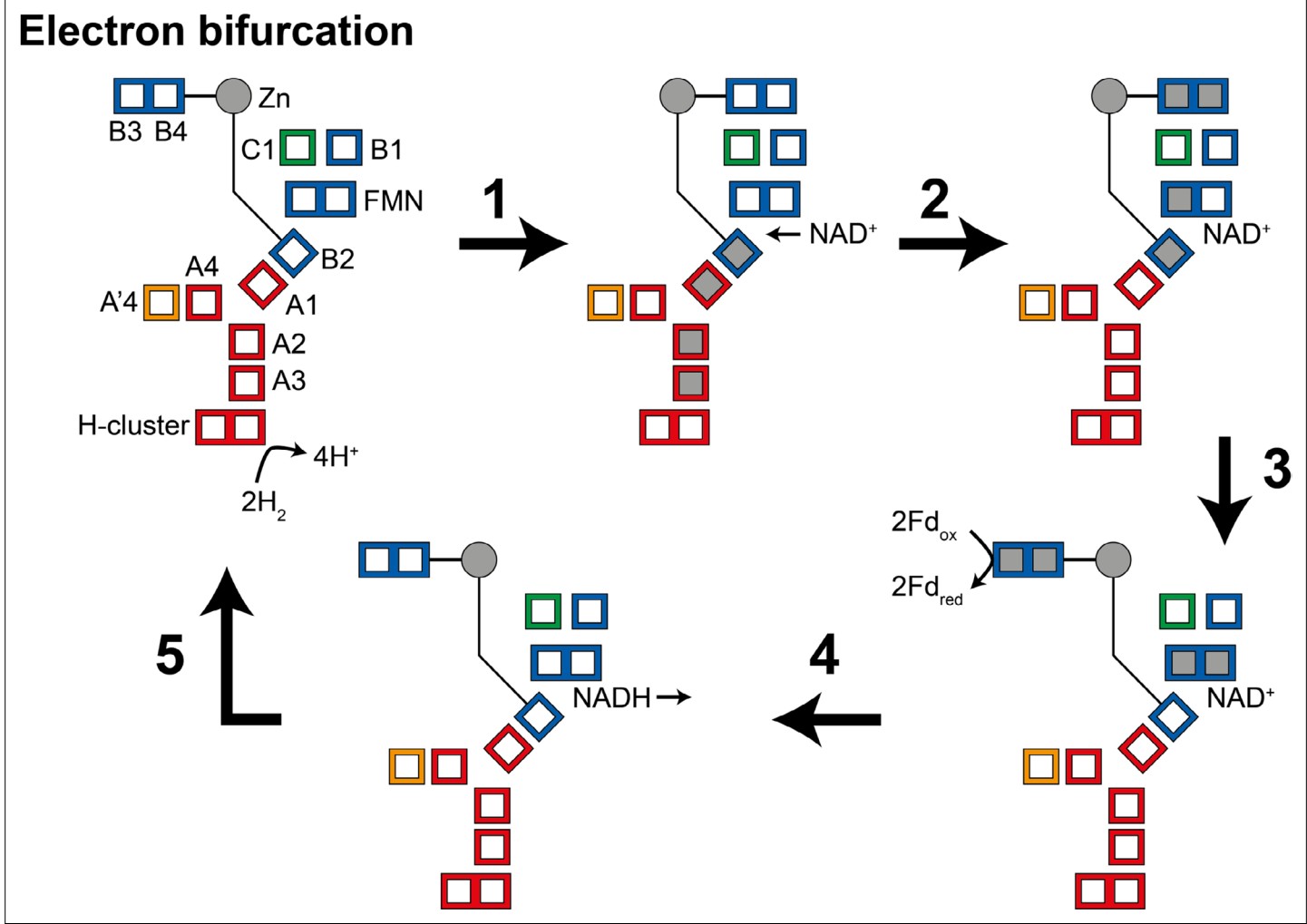

**Figure 5.** Illustration of a possible mechanism of electron transfer in HydABC during electron bifurcation. (1) Electrons generated by oxidation of $H_2$ at the H-cluster travel down the core electron transfer pathway to the B2 cluster but not to the flavin mononucleotide (FMN). Reduction of the B2 cluster triggers bridge movement allowing the B3/B4 clusters to get close to the B1 and C1 clusters. (2) $NAD^+$ binding stabilizes the $FMN^{•-}$ radical allowing electron transfer to the FMN, then to the B1/C1 clusters, and finally to the B3 and B4 clusters. (3) The bridge domain then returns to the closed position allowing reduction of ferredoxin. (4) Domain movement triggers the FMN to get fully reduced to the $FMNH^-$ state, which can then reduce $NAD^+$ to NADH. (5) NADH is released and the enzyme returns to its initial state. The reverse, electron confurcation, direction (NADH and reduced ferredoxin are used to produce $H_2$) is described in *Figure 5—figure supplement 1*. Color code: red regions are in HydA, orange regions are in HydA', blue regions are in HydB, and green regions are in HydC. The gray circle indicates the Zn site. Gray squares represent the location of electrons.

The online version of this article includes the following figure supplement(s) for figure 5:

**Figure supplement 1.** Illustration of a possible mechanism of electron transfer in HydABC during electron confurcation.

the C1 and B1 clusters. $FMN^{•-}$ cannot reduce $NAD^+$ as the $NAD^•$ radical is very unstable but $FMN^{•-}$ can reduce the C1 cluster, which in turn reduces the B3 and B4 clusters via the B1 cluster. (3) Fd-binding triggers a conformational change, moving the B3 and B4 clusters away from the C1 and B1 clusters and closer to the Fd-binding site. This conformational change also alters the potentials of the FMN so that $FMN^{•-}$ can be reduced to $FMNH^-$ by the B2 cluster. (4) The final stage is hydride transfer from $FMNH^-$ to $NAD^+$ to make NADH and reduction of Fd by the B3 and B4 clusters. HydABC is known to also function in the reverse, electron confurcating, direction where electrons from NADH and reduced ferredoxin are channeled toward the H-cluster and used to reduce $H^+$ to $H_2$. In the electron confurcating direction (*Figure 5—figure supplement 1*): (1) ferredoxin reduces the B3 and B4 clusters while the bridge is in the closed state. NADH binds and transfers a hydride to the FMN to make $FMNH^-$. (2) $FMNH^-$ transfers an electron to the B2 cluster triggering the bridge to open allowing it to move close enough to transfer electrons to the B1 and C1 clusters. (3) Electrons are transferred to the H-cluster via

the C1, B1, A1, A2, and A3 clusters. (4) NAD$^+$ dissociation triggers the bridge to close again and the potentials of the FMN to change such that FMN$^{\bullet-}$ transfers its electron to the core electron transfer pathway. The electrons in the core pathway can reduce 4H$^+$ to 2H$_2$.

The mechanism described above is highly speculative at present but does make some important predictions. We expect that NADH binding to HydABC would generate a stable FMN$^{\bullet-}$ radical, leading to the reduction of a single [4Fe–4S] cluster (B2), and triggering the HydB-CT domain to open. Meanwhile, ferredoxin is expected to reduce the B3 and B4 clusters only, and reduction of C1/B1 and all clusters in HydA will only be observed upon the addition of both NADH and ferredoxin. Additionally, H$_2$ oxidation will reduce clusters in HydA as well as cluster B2, leading to an opening of the HydB-CT domain. H$_2$ and NAD$^+$ would be expected to lead to the reduction of C1, B1, B3, and B4 as well as the formation of an FMN$^{\bullet-}$ radical.

A similar mechanistic proposal was made by *Feng et al., 2022* to explain electron bifurcation in the related [NiFe] hydrogenase (HydABCSL) from *A. mobile*. HydA, B, and C in *A. mobile* are homologous to HydA, B, and C in *T. maritima*, respectively, however, HydA in *A. mobile* lacks the H-cluster and instead the enzyme contains HydS and L, which form the [NiFe] hydrogenase unit. The fact that both enzymes bifurcate electrons, yet do not both contain the H-cluster, further supports the idea that the H-cluster is not the site of electron bifurcation in *Tm*HydABC. Otherwise, the structures of the HydABC units in both enzymes are very similar. However, it was proposed that instead of a zinc site *Am*HydB contains an additional [2Fe–2S] cluster, which allows electron transfer between the site of ferredoxin oxidation in the B3/B4 clusters and the [2Fe–2S] cluster in *Am*HydC. The latter was also suggested to be located in a mobile domain and that conformational changes are triggered by events at the FMN site. However, the authors did not consider in detail how nucleotide binding or changes in the FMN redox potentials could be coupled to conformational changes. While the two mechanistic proposals differ in the details, they both consider the FMN and unique arrangement of metallocofactors around it to be crucial components for electron bifurcation.

In summary, our cryo-EM structure reveals essential information on the arrangement of cofactors and active sites within *T. maritima* HydABC, including interprotomer electronic wiring. Using symmetry expansion, we have also observed two conformations of the HydB-CT domain, a domain that is unique to and conserved in bifurcating hydrogenases, consistent with mechanistically relevant conformational changes. These structural revelations open up new avenues for exploring the ways in which flavins can bifurcate electrons. Such a mechanism may also be operative in other enzymes homologous to HydABC. By resolving these crucial structural details, the mechanism of bifurcation can be further investigated by studying the role of the FMN and the HydB C-terminal domain using site-directed mutagenesis coupled with kinetic and spectroscopic studies. Further structural studies are also underway with holo-HydABC to investigate the precise structural details of the H-cluster, the effects of reduction by H$_2$, as well as the conformational changes induced by nucleotide and ferredoxin binding. These findings will then be correlated with spectroscopic and functional information to provide a detailed understanding of the mechanism of electron bifurcation in this interesting enzyme.

## Methods

### Key resources table

| Reagent type (species) or resource | Designation | Source or reference | Identifiers | Additional information |
|---|---|---|---|---|
| Strain, strain background (*Escherichia coli*) | BL21(DE3)Δ*iscR*/pASK-IBA17plus/*hydabc* | *Chongdar et al., 2020* | n/a | A genetically modified strain of *E. coli* containing a kanamycin resistance cassette inserted in the *iscR* gene and transformed with a pASK-IBA17plus plasmid containing the *hydabc* protein-coding DNA sequence |
| Chemical compound, drug | Strep-Tactin Superflow high capacity resin | IBA-life sciences | 2-1208-025 | Used for purification of *Tm*HydABC |
| Software, algorithm | RELION-3.1 | *Zivanov et al., 2019* | n/a | Image processing |
| Software, algorithm | WinCoot | *Emsley et al., 2010* | n/a | Modeling |

*Continued on next page*

*Continued*

| Reagent type (species) or resource | Designation | Source or reference | Identifiers | Additional information |
|---|---|---|---|---|
| Software, algorithm | Phenix | *Liebschner et al., 2019* | n/a | Model refinement |
| Software, algorithm | ChimeraX 1.1 | *Pettersen et al., 2021* | n/a | Used to visualize maps and models and to make the figures in this paper |
| Software, algorithm | EasySpin 5.2.35 | *Stoll and Schweiger, 2006* | n/a | Used to simulate EPR spectra |
| Other | UltrAuFoil R 1.2/1.3 Gold foil on Gold 300 mesh grid | Quantifoil Micro Tools GmbH | n/a | Used to prepare cryo-EM grids |

## Protein expression and purification

Previously, HydABC was expressed heterologously in *E. coli* and purified under anaerobic conditions, generating an 'apo' enzyme, containing all of the [2Fe–2S] and [4Fe–4S] clusters, but lacking the $[2Fe]_H$ subcluster of the H-cluster in HydA (*Chongdar et al., 2020*; *Kuchenreuther et al., 2010*). The H-cluster was then reconstituted using a synthetic $[2Fe]_H$ precursor (*Chongdar et al., 2020*; *Berggren et al., 2013*; *Esselborn et al., 2013*). The H-cluster of [FeFe] hydrogenases (including HydABC) is highly sensitive to $O_2$ (*Swanson et al., 2015*). Additionally, only minor structural differences are observed upon incorporation of the [2Fe] subcluster (*Esselborn et al., 2016*). As our grid preparation was only possible under air and our main interest was in the structural characterization of the electron transfer pathways, we decided to focus on the 'apo' enzyme. Previous studies with [FeFe] hydrogenase (*Cp*I) from *C. pasteurianum* showed that the 'apo' and 'holo' enzymes have identical structures (*Esselborn et al., 2016*). For this work, HydABC and HydB were produced heterologously in *E. coli* BL21(DE3) *ΔiscR* cells under anaerobic growth conditions and purified in an anaerobic glovebox (Coy, 2% $H_2$ in $N_2$) using Streptactin (IBA) affinity chromatography and size-exclusion chromatography (GE Healthcare) as previously described (*Chongdar et al., 2020*). For these studies, we did not incorporate the $[2Fe]_H$ subcluster to form the holo-enzyme. Sample purity and quality were checked by sodium dodecyl sulfate–polyacrylamide gel electrophoresis and UV–vis spectrophotometry. Samples in 10 mM Tris–HCl, 150 mM NaCl, pH 8 were frozen at −80°C until further use.

## Inductively coupled plasma mass spectrometry

For inductively coupled plasma mass spectrometry (ICP-MS), a sample of the HydB subunit, buffer exchanged into 10 mM MOPS pH 7 and concentrated to 621 µM, and a sample of 10 mM MOPS pH 7 were measured by Mikroanalytisches Laboratorium Kolbe (https://www.mikro-lab.de/). The samples were digested using a CEM Model MARS6 microwave digestion unit and measured on an Agilent Model 7900 ICP-MS.

## Grid preparation and imaging

1.2/1.3 UltrAuFoil grids were glow discharged (PELCO easiGlow) for 90 s on each side using atmospheric gas before mounting in Vitrobot (model IV) tweezers (Thermo Fisher Scientific). We prepared grids with minimal exposure to air using anaerobically frozen aliquots of HydABC. These were individually defrosted and used. In this manner, HydABC was exposed to the air for a few seconds. The enzyme (without the [FeFe] site) seems to be stable under air for at least a few hours, determined as there were no visible spectral changes when the enzyme solution was exposed to air. Individual HydABC aliquots were defrosted and 2.5 µl immediately placed onto the grid, blotted, and plunged into liquid ethane. 12 grids were prepared, varying blot time from 2 to 4 s with 0.75–1.5 mg ml$^{-1}$ protein; blot force parameter was constant at −5. Following screening to optimize protein concentration and blotting parameters, cryo-grids could be consistently prepared with densely packed but non-aggregated particles where it was possible to see several different views of HydABC by eye. Following screening, a grid at 1 mg ml$^{-1}$ protein concentration was selected for data collection on a Titan Krios microscope operated at 300 kV with a K2 detector and energy filter. The energy filter was

set to a 20 eV window. Three exposures were collected per hole, and the autofocus routine was run every 10 µm. AutoCTF was used to correct for astigmatism and coma. 4790 movies of 48 frames each were collected. The total fluence was 57 electrons / $\text{Å}^2$.

## Image processing

The Relion pipeline was used for all image processing. Whole micrograph motion correction and damage weighting were performed using the implementation of MotionCor2 in Relion (*Zivanov et al., 2018*). Initial CTF values were determined with CTFFIND4 (*Rohou and Grigorieff, 2015*) and particles were picked using a low resolution (≈10 Å) preliminary dataset that was previously collected (not described here). The early stages of 2D and 3D classification used images with the original pixel size downsampled from 0.85 to 3.4 Å/pixel. Reference-free 2D classification was performed to classify the particles (*Figure 1—figure supplement 3*) and remove broken particles that are most likely denatured at the air–water interface, common to most cryo-EM projects (*Noble et al., 2018*). It was clear there were large particles that had four lobes consistent with a tetramer of trimers and smaller particles, with high-resolution features (*Figure 1—figure supplement 3*). Any classes that showed high-resolution features in the 2D class averages were selected for coarse 3D classification, which effectively cleaned the dataset to only the tetramer of trimer particles, consistent with the gel filtration profile of the preparation. An initial model was generated in Relion and coarse 3D classification (7.5° sampling) without symmetry being enforced was used to remove broken particles. Docking in the related structure of subunits Nqo1, Nqo2, and Nqo3 of complex I from *T. thermophilus* (*Baradaran et al., 2013*) showed that the particles had D2 symmetry, consistent with a tetramer of trimers Hyd(ABC)$_4$ arrangement. The particles were reextracted with the original pixel size of 0.85 Å/pixel and 3D autorefinement of these particles resulted in a 2.5-Å resolution structure when D2 symmetry was applied. To further improve the resolution, anisotropic magnification, trefoil, and fourth-order aberration parameters were refined; with astigmatism and defocus being fitted on a per-particle basis (*Zivanov et al., 2020*). Bayesian polishing was also performed (*Zivanov et al., 2019*). The map displayed the features expected at such a resolution, with rotamers of many side chains being clear and water molecules being visible in well-resolved regions. Refinement resulted in a 2.3-Å resolution structure when D2 symmetry was applied (*Figure 1—figure supplement 3*). The final calibrated pixel size was 0.824 Å.

To investigate the blurred bridging regions, symmetry expansion was used to separate the different conformations into classes. Here, particles with symmetry are transformed so that each symmetry-related subparticle is overlaid; a mask is then applied so each subparticle can be treated independently for classification and refinement (*Ilca et al., 2015*). The high-resolution D2 refinement was used as a starting point. As each lobe appeared independent of the others, symmetry expansion with D2 symmetry to match the core was attempted to separate the different conformations into classes but this was unsuccessful, resulting in maps no clearer than the first. However, when the same process was repeated using C2 symmetry much better results were found. To achieve this, the relion_symmetry_expand command was used to apply a C2 symmetry operator to the particles in the refined.star file. A 20 Å low-pass filtered mask, generated from fitted atomic coordinates and expanded by 20 pixels with 6 pixels soft edge, was then applied to half of the complex containing two tightly connected HydABC protomers with a complete and connected electron transfer network. A clear bridging density was found to exist between two HydBC lobes in a subset of Hyd(ABC)$_2$ particles (total 39.1%). A tighter mask was then created that included exclusively the two 'bridges' densities in the Hyd(ABC)$_2$ unit (20 Å low-pass filter, 6 pixels soft edge), allowing a better 3D classification without losing any signal in the 'bridges'. The resulting 'bridged' classes (bridge backward and forward) were refined with C1 symmetry applying a 6-pixel soft edge mask that included the Hyd(ABC)$_2$ unit with two bridges, reaching a resolution of 2.8 Å for both the classes. In this subset, half of the particles had the bridge forward with respect to the rest of the enzyme (i.e., bridging from A to B') and the other had the bridge backward (i.e., bridging from A' to B), but none showed both the bridges with clear density (*Figure 4—figure supplement 1A*). The bridge is formed by the C-terminus of HydA (containing one [2Fe–2S] cluster) from one protomer and the C-terminus of HydB (containing two [4Fe–4S] clusters) from the neighboring protomer, thereby breaking the rotational symmetry between the two bridged lobes.

To explore the location of the HydB in the non-bridged class, a mask was created around the suspected area and used for classification and refinement (*Figure 4—figure supplement 1B*). The

improved map allowed an improved mask to be created for a final round of classification and refinement. The resulting map density is of insufficient quality for ab initio model building, but the strong FeS signals allowed the HydB CT-domain to be docked in place (*Figure 4D*).

## Model building and validation

WinCoot (*Emsley et al., 2010*) and Phenix (*Liebschner et al., 2019*) were used for model building and validation, and ChimeraX (*Pettersen et al., 2021*) was used for visualization and figure generation. We used a homology model generated based on bacterial complex I (*Baradaran et al., 2013*) discussed in our recent paper on HydABC (*Chongdar et al., 2020*) as a starting point for model building. Here, the Nqo3 subunit of complex I is related to HydA, Nqo1 to HydB, and Nqo2 to HydC. The map density was sufficiently strong to allow ab initio building of the non-conserved regions of HydA and HydB in the well-resolved parts of the D2 map, however, without further classification, many parts of HydB and HydC were poorly resolved. Model refinement was performed using Phenix real-space refinement. Phenix now automatically recognizes the ligation between FeS clusters and cysteines, so it is no longer necessary to manually define these restraints or to provide the correct definition of the FeS geometry (*Moriarty and Adams, 2019*).

The 'bridge' is formed from 91 residues of the CT of HydA and 61 residues of the CT of HydB. The HydA CT 'bridge' domain has homology with the CT of HndA from the NADP-reducing hydrogenase complex in *Desulfovibrio fructosovorans* (*Nouailler et al., 2006*) and 82 CT residues of *T. maritima* HydC. The HydB CT 'bridge' domain has homology with bacterial 2×[4Fe–4S] ferredoxin domains. In both cases, Phyre2 was used to build a homology model from this information, which was further built into the density, combined with the model for the rest of the complex built from the D2 map and refined (*Kelley et al., 2015*).

## EPR spectroscopy

A 0.2 ml, 0.2 mM sample of the HydB subunit in 100 mM Tris–HCl, 150 mM NaCl, pH 8 reduced with 10 mM sodium dithionite was transferred to a quartz 4 mm (o.d.) EPR tube and frozen in liquid nitrogen. X-band EPR spectra were recorded on a Bruker ELEXSYS E500 CW X-band EPR spectrometer. The temperature of the sample was controlled using an Oxford Instruments ESR900 helium flow cryostat connected to an ITC503 temperature controller. The measurement parameters were: microwave frequency 9.64 GHz, time constant 81.92 ms, conversion time 81.92 ms, and modulation frequency 100 kHz. The microwave power and temperature were varied between measurements and are indicated in the figure legends. All spectra were analyzed with home-written scripts in MATLAB. Spectral simulations were performed using the EasySpin package (*Stoll and Schweiger, 2006*).

## Acknowledgements

This work benefited from access to the Astbury Biostructure Laboratory, an Instruct-ERIC center. Financial support was provided by Instruct-ERIC (PID 11666). We are grateful to Svetomir Tzokov, Charlotte Scarff, and Rebecca Thompson for assistance with data collection and Nigel Moriarty for assistance with FeS ligation during model building. We used computational resources provided by the Viking Cluster at the University of York and are grateful to the Research Computing team and Huw Jenkins for assistance with computing. We are grateful to Hannah Bridges, Laure Decamps, and Patrícia Rodríguez Maciá for critical evaluation of the manuscript. JAB and NC acknowledge funding from the DFG SPP 1927 'Iron–Sulfur for Life' project (Project No. BI 2198/1-1). The work was supported by the Max Planck Society (JAB, NC, and WL), and in part a UKRI Future Leader Fellowship (JNB; MR/T040742/1) and JSPS KAKENHI (grant number JP20H03215 [HO]). This manuscript is dedicated to Yvonne Brandenburger.

## Additional information

### Funding

| Funder | Grant reference number | Author |
|---|---|---|
| Deutsche Forschungsgemeinschaft | BI 2198/1-1 | Nipa Chongdar |
| UK Research and Innovation | MR/T040742/1 | James N Blaza |
| Japan Society for the Promotion of Science | JP20H03215 | Hideaki Ogata |
| Max-Planck-Gesellschaft | | Nipa Chongdar |

The funders had no role in study design, data collection, and interpretation, or the decision to submit the work for publication.

### Author contributions

Chris Furlan, Data curation, Formal analysis, Validation, Visualization, Methodology, Writing – original draft, Writing – review and editing; Nipa Chongdar, Conceptualization, Resources, Investigation, Writing – original draft, Writing – review and editing; Pooja Gupta, Resources, Data curation, Formal analysis, Validation, Investigation, Visualization, Methodology, Writing – review and editing; Wolfgang Lubitz, Conceptualization, Resources, Supervision, Funding acquisition, Writing – review and editing; Hideaki Ogata, Conceptualization, Resources, Funding acquisition, Project administration, Writing – review and editing; James N Blaza, Conceptualization, Resources, Data curation, Software, Formal analysis, Supervision, Funding acquisition, Validation, Investigation, Visualization, Methodology, Writing – original draft, Project administration, Writing – review and editing; James A Birrell, Conceptualization, Resources, Supervision, Funding acquisition, Writing – original draft, Project administration, Writing – review and editing

### Author ORCIDs

James N Blaza http://orcid.org/0000-0001-5420-2116
James A Birrell http://orcid.org/0000-0002-0939-0573

### Decision letter and Author response

Decision letter https://doi.org/10.7554/eLife.79361.sa1
Author response https://doi.org/10.7554/eLife.79361.sa2

## Additional files

### Supplementary files

• Supplementary file 1. Tables summarizing the structural features of HydABC from *Thermotoga maritima* and the cryo-EM data collection and refinement statistics. (a) Table summarizing the buried surface area (in $Å^2$), number of salt bridges and number of hydrogen (H-)bonds between the various interacting partners in the heterododecameric Hyd(ABC)$_4$ complex. (b) Table summarizing the cryo-EM data collection and refinement statistics of the four structural models presented in this work: (1) the D2 tetramer of trimers model (PDB ID: 7P5H), (2) the bridge closed forward model (PDB ID: 7P8N), (3) the bridge closed reverse model (PDB ID: 7P91), and (4) the open bridge model (PDB ID: 7P92).

• MDAR checklist

### Data availability

Protein databank (PDB) files for the four model presented in this manuscript are available at https://www.rcsb.org/ under PDB ID 7P5H D2 tetramer, 7P8N (Bridge closed forward), 7P91 (Bridge closed reverse), and 7P92 (Open bridge). Cryo-EM maps are available at https://www.ebi.ac.uk/pdbe/emdb/ under EMD-13199, EMD-13254, EMD-13257 and EMD-13258 . All other data are available in the main text or the supplementary materials.

The following datasets were generated:

| Author(s) | Year | Dataset title | Dataset URL | Database and Identifier |
|---|---|---|---|---|
| Furlan C, Chongdar N, Gupta P, Lubitz W, Ogata H, Blaza JN, Birrell JA | 2022 | TmHydABC- D2 map | https://www.rcsb.org/structure/7P5H | RCSB Protein Data Bank, 7P5H |
| Furlan C, Chongdar N, Gupta P, Lubitz W, Ogata H, Blaza JN, Birrell JA | 2022 | TmHydABC- T. maritima hydrogenase with bridge closed | https://www.rcsb.org/structure/7P8N | RCSB Protein Data Bank, 7P8N |
| Furlan C, Chongdar N, Gupta P, Lubitz W, Ogata H, Blaza JN, Birrell JA | 2022 | TmHydABC- T. maritima bifurcating hydrogenase with bridge domain closed | https://www.rcsb.org/structure/7P91 | RCSB Protein Data Bank, 7P91 |
| Furlan C, Chongdar N, Gupta P, Lubitz W, Ogata H, Blaza JN, Birrell JA | 2022 | TmHydABC- T. maritima bifurcating hydrogenase with bridge domain up | https://www.rcsb.org/structure/7P92 | RCSB Protein Data Bank, 7P92 |

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
