## [Editor Report]

This paper describes a high resolution cryo-EM structure of an [FeFe] hydrogenase purported to operate via an electron bifurcating mechanism. The study aims to resolve a controversy regarding the site of bifurcation through structural characterization of the enzyme complex. The authors propose a mechanism for electron transfer in which conformational changes and cofactor binding events modulate the properties of the pathway.

---

## [Decision Letter]

**Decision letter after peer review:**

Thank you for submitting your article "Structural insight on the mechanism of an electron-bifurcating [FeFe] hydrogenase" for consideration by *eLife*. Your article has been reviewed by 2 peer reviewers, one of whom is a member of our Board of Reviewing Editors, and the evaluation has been overseen by Volker Dötsch as the Senior Editor. The reviewers have opted to remain anonymous.

Essential revisions:

1) The authors must address the lack of essential cofactors in their structure more explicitly. Ideally, a structure with the active site H-cluster would be reported. But, at minimum, characterization of the current apo preparation is required to show that the cofactor content is the same as active enzyme preparations containing the H-cluster. Additionally, the manuscript and figures should be revised to comment on the decision to characterize this inactive version of the enzyme and to better highlight the H-cluster omission and the associated drawbacks of using this structure to evaluate reactivity.

2) The manuscript should also be revised to state more explicitly the experimental results that form the basis for an electron bifurcation mechanism in this system.

3) The authors should also include a more extended discussion of the similarities between their conclusions about the role of an FMN cofactor in electron bifurcation and an analogous proposal put forth by Adams et al. in their recently reported cryo-EM characterization of a structurally related NiFe hydrogenase.

*Reviewer #1 (Recommendations for the authors):*

The manuscript reports a high-resolution structure of a sought-after enzyme target, revealing the location and spatial arrangement of many of its key cofactors. The authors use this information to propose a detailed mechanism for shuttling reducing equivalents to the hydrogenase active site. In this pathway, they implicate the FMN cofactor in an important bifurcation step. The mechanism proposal allows the authors to make several testable predictions that could form the basis for future studies. These are key strengths of the manuscript. The manuscript could be further strengthened by including a more transparent discussion of the omission of the active site H-cluster cofactor. The limitations of using a structure that lacks the full complement of cofactors – particularly in ruling out an existing hypothesis about the role of the H-cluster in electron bifurcation – are not clear in the current manuscript. Another weakness is the lack of description of the experimental basis for use of an electron bifurcation mechanism by this enzyme. Finally, the manuscript could be improved by including a more complete discussion of a study published earlier this year that reports a very similar mechanism in a structurally related NiFe hydrogenase.

The manuscript includes a proposal for an electron transfer pathway in which the sole FMN in HydB is the bifurcation site. This discussion includes several testable hypotheses that are explicitly stated in the manuscript. The work would be strengthened by including some experimental validation of these ideas.

The primary goal of this study is to understand the mechanism of electron transfer, specifically to identify the site of a proposed electron bifurcation step. The manuscript describes two hypotheses for the cofactor implicated in electron bifurcation – a second flavin cofactor or the H-cluster involved in proton reduction in the active site of the hydrogenase. The structure rules out the first hypothesis because it does not provide evidence for a second flavin. But the structure reported here is of an apo form of the hydrogenase generated by heterologous overexpression, meaning that it lacks the H-cluster. Therefore, it is not clear how the second hypothesis can be dismissed at this stage. The omission of the H-cluster should be discussed more transparently in the manuscript. As it stands, the lack of this key active site cofactor could be missed because it is only briefly mentioned twice in the Results section. And, more importantly, many of the figures show the H-cluster as though it is present in the cryo-EM model. If this component of the active site is modeled based on its presumed location – rather than its observation in the cryo-EM map – then this distinction should be indicated as such in the figures. And the limitations of interpreting the structure due to the missing H-cluster should be more explicitly discussed. In the current draft, this important detail could be easily missed by the reader. It would be ideal to either obtain a structure of the protein reconstituted with the H-cluster or to further characterize the apo preparation to show that its cofactor content, properties, etc are not different from the active version containing the H-cluster.

The premise of the paper relies heavily on the electron bifurcation chemistry proposed to be important in this enzyme. But the experimental basis for this phenomenon in this hydrogenase homolog is not clearly explained. It is stated on page 4, line 71, that the mechanism of HydABC is likely a flavin-based electron bifurcation one – but the experimental results that support this conclusion are not explained. It is important to understand the basis for this phenomenon in this enzyme given that the entire manuscript is centered on this idea.

Earlier this year Adams et al. published a series of cryo-EM structures of a structurally related NiFe hydrogenase that contains an HydABC module that is similar to the enzyme core reported here. This manuscript is cited twice and discussed briefly – mostly to highlight minor structural/cofactor differences. However, the Adams study seems to reach a similar conclusion – that the sole flavin cofactor is the site of bifurcation and that conformational changes modulate the properties of the FMN and other cofactors linked to it in the pathway. The common features of the two systems would seem to warrant a more extended discussion of the similarities.

Throughout the manuscript, the authors use unconventional terminology that is difficult to understand. In the abstract, on line 27, the two heterotrimers are described as "electrically connected." I don't understand what this term means – but the authors use it repeatedly in the main body of the manuscript. In some cases, I would guess that the intended meaning refers to two cofactors that are close enough for single step protein-mediated electron transfer (within >15-20 Å) – but it is difficult to tell because of the imprecision of the language.

In the discussion, on page 12 lines 309-311, the authors refer to electron transfer between "redox couples." This description also lacks precision. The "couple" does not transfer electrons. Electron transfer occurs between the cofactors. This section should be rewritten accordingly.

Figure 5 is challenging to understand in its current form. The gray shading of certain boxes is not described in the figure or the caption. The text is very small. And the term "electron confurcation" is not defined well in the manuscript text.

*Reviewer #2 (Recommendations for the authors):*

The manuscript "Structural insight on the mechanism of an electron-bifurcating [FeFe] hydrogenase" by Dr Blaza, Dr Birrell, and co-workers reports the first electron cryo-microscopy structure of a multi-subunit [FeFe]-hydrogenase. The enzyme is an electron-bifurcating hydrogenase that synergistically oxidises NADH and ferredoxin releasing hydrogen (H2). The structure reveals that the enzyme is composed of a dodecamer formed by a tetramer of the three subunits HydA, HydB, and HydC. A closer inspection of the 3D arrangement of the redox centre (several FeS clusters and FMN) reveals that the electron transfer pathway is not linear, but several branching points exist, as well as opportunities for exchanging electrons between adjacent HydABC protomers. A structural zinc site is identified and a conformational change is proposed to play a key role in the electron bifurcation mechanism.

Strengths.

Structural characterisation of [FeFe]-hydrogenases is very limited, due to historic limitations in producing and purifying these O2-sensitive enzymes. This paper sets a milestone by revealing the structure of a new enzyme from this class.

The methodology is well suited to studying the electron bifurcating HydABC enzyme from *Thermotoga maritima*. Extensive references to the literature and comparison to other bifurcating enzymes make the paper very well presented.

The results contribute to understanding the electron bifurcation mechanism, which spans far beyond [FeFe]-hydrogenases, and is crucial in energy conservation.

Weaknesses.

The main weakness is that the manuscript does not explore the structure in the presence of all cofactors, particularly the H-cluster (active site of HydA), NAD+/NADH (redox partner of HydB) and ferredoxin (redox partner of HydB and/or HydC).

As outlined above, this piece of research is really good and should be accepted for publication. However, some points should be addressed beforehand.

The main concern is that many of the mechanistic discussions could potentially be strengthened by exploring further the enzyme structure (and its conformational changes) in the presence of its cofactors, particularly NAD+/NADH and ferredoxin. I fully appreciate that this work may be complicated by the binding of these cofactors being labile and being driven by specific redox state(s) of the enzyme clusters. I also appreciate that this would require a large amount of additional work. For this reason, I think that it is not appropriate to request that additional experiments are performed now, but it would be ideal if the authors expand the discussion to comment further. For example, would these experiments be feasible with cryo-EM? Have they been attempted without success? Are these planned for future work? If not, what alternative techniques can be used to prove/disprove the proposed mechanism?

The absence of the [FeFe] subcluster of the H-cluster should be discussed more clearly. The authors should discuss in the manuscript if this was omitted because the sample is exposed to oxygen during cryo-EM processing? Is this unavoidable? If this decision was based on other factors, these should also be discussed in the manuscript.

---

## [Author Response]

Essential revisions:1) The authors must address the lack of essential cofactors in their structure more explicitly. Ideally, a structure with the active site H-cluster would be reported. But, at minimum, characterization of the current apo preparation is required to show that the cofactor content is the same as active enzyme preparations containing the H-cluster. Additionally, the manuscript and figures should be revised to comment on the decision to characterize this inactive version of the enzyme and to better highlight the H-cluster omission and the associated drawbacks of using this structure to evaluate reactivity.

We chose to characterize the enzyme lacking the [2Fe]_H_ subcluster of the H-cluster, but containing all other cofactors, because this allowed us to prepare the cryoEM grids under air, and we hypothesized that the absence of the [2Fe]_H_ subcluster would not substantially affect the global protein structure, as this was already well-known for the homologous single subunit hydrogenase (*Cp*I) from *Clostridium pasteurianum* (see Esselborn et al., *Chem. Sci.*, 2016). In fact, comparison of our cryoEM structure with the structure of CpI, shows that the structure of the region around the H-cluster is almost identical in both enzymes, supporting our hypothesis that the [2Fe]_H_ cofactor insertion does not substantially change the protein structure. We also observe all other predicted cofactors in the structure and we showed in a previous publication that the Fe content of the apo-enzyme (35 Fe/HydABC) matched the expected Fe content (36 Fe/HydABC) and was slightly higher than previous measurements on the native enzyme from *Thermotoga maritima* (32 Fe/HydABC). We apologise that this was not made clear enough in the original version of the manuscript. Now in the revised version of the manuscript we have modified the beginning of the Results section on Page 3 as follows: “In our previous work, it was shown that this preparation contains all the redox cofactors of the native HydABC enzyme except for the [2Fe] subcluster of the hydrogenase active site (H-cluster), which *E. coli* is unable to synthesize. In particular, Fe quantitation measurements of the heterologously produced enzyme agreed with the expected number of iron-sulfur clusters based on sequence analysis, and were even higher than those from the native enzyme (Verhagen BBA 1999). Furthermore, EPR spectra of the reduced apo- and reduced holo-HydABC (where the H-cluster is EPR-silent) were identical to each other and the same as those from the native enzyme (Verhagen BBA 1999). A drawback of using this apo-HydABC preparation is that we cannot observe how the structure is affected by reduction by H_2_.”

We have also modified Figures 1 to 4 to replace “H-cluster” and “H” with “[4Fe-4S]_H_, and we have added the following sentence to the captions of Figures 1, 2 and 4: “Note that our structure is of the apo-HydABC and lacks the [2Fe]H subcluster of the H-cluster.”

We also added the following sentence on Page 14 of the Results: “Importantly, the structure around the H-cluster is highly conserved between CpI and HydABC with only very small deviations in the positions of serveral conserved side-chains (Figure 3—figure supplement 1).” and added a supplementary figure (Figure 3—figure supplement 1).

Finally, we have added a sentence at the end of the discussion on Page 22 outlining the potential novel information that will be provided by solving additional structures of the active enzyme containing [2Fe]_H_: “Further structural studies are also underway with holo-HydABC to investigate the precise structural details of the H-cluster, the effects of reduction by H_2_, as well as the conformational changes induced by nucleotide and ferredoxin binding. These findings will then be correlated with spectroscopic and functional information to provide a detailed understanding of the mechanism of electron bifurcation in this interesting enzyme.”

2) The manuscript should also be revised to state more explicitly the experimental results that form the basis for an electron bifurcation mechanism in this system.

We acknowledge that this was a limitation in the original version of the manuscript and have done our best to revise the manuscript to highlight the information obtained from our structures that form the basis for our mechanistic proposal. We acknowledge that the proposal is still largely speculative but this will form the basis for planning future experiments and may also help other researchers working on similar enzymes to consider whether our mechanistic proposal could also help explain their observations. As such we feel that putting our hypotheses out into the community will be a useful exercise.

On Page 20 of the discussion we added : “Our first structure reveals that the FMN is located at a branch point connecting the core electron transfer pathway from the H-cluster and the additional iron-sulfur clusters B1 and C1, while our additional structures reveal that the FMN is close to a zinc site and a mobile iron-sulfur cluster domain, all indicating that it is ideally located for behaving as an electron bifurcation center.”

3) The authors should also include a more extended discussion of the similarities between their conclusions about the role of an FMN cofactor in electron bifurcation and an analogous proposal put forth by Adams et al. in their recently reported cryo-EM characterization of a structurally related NiFe hydrogenase.

Indeed, we agree that a more detailed discussion between our structure/mechanism and that recently published by the Adams group would enhance the interest of our paper. As such, we have added the following sentence on Page 21: “A similar mechanistic proposal was made by Feng et al. (Feng Sci Adv 2022) to explain electron bifurcation in the related [NiFe] hydrogenase (HydABCSL) from A. mobile. HydA, B and C in A. mobile are homologous to HydA, B and C in T. maritima, respectively, however, HydA in A. mobile lacks the H-cluster and instead the enzyme contains HydS and L, which form the [NiFe] hydrogenase unit. The fact that both enzymes bifurcate electrons, yet do not both contain the H-cluster, further supports the idea that the H-cluster is not the site of electron bifurcation in TmHydABC. Otherwise, the structures of the HydABC units in both enzymes are very similar. However, it was proposed that instead of a zinc site AmHydB contains an additional [2Fe-2S] cluster, which allows electron transfer between the site of ferredoxin oxidation in the B3/B4 clusters and the [2Fe-2S] cluster in AmHydC. The latter was also suggested to be located in a mobile domain and that conformational changes are triggered by events at the FMN site. However, the authors did not consider in detail how nucleotide binding or changes in the FMN redox potentials could be coupled to conformational changes. While the two mechanistic proposals differ in the details, they both consider the FMN and unique arrangement of metallocofactors around it to be crucial components for electron bifurcation.”

Reviewer #1 (Recommendations for the authors):The manuscript includes a proposal for an electron transfer pathway in which the sole FMN in HydB is the bifurcation site. This discussion includes several testable hypotheses that are explicitly stated in the manuscript. The work would be strengthened by including some experimental validation of these ideas.The primary goal of this study is to understand the mechanism of electron transfer, specifically to identify the site of a proposed electron bifurcation step. The manuscript describes two hypotheses for the cofactor implicated in electron bifurcation – a second flavin cofactor or the H-cluster involved in proton reduction in the active site of the hydrogenase. The structure rules out the first hypothesis because it does not provide evidence for a second flavin. But the structure reported here is of an apo form of the hydrogenase generated by heterologous overexpression, meaning that it lacks the H-cluster. Therefore, it is not clear how the second hypothesis can be dismissed at this stage. The omission of the H-cluster should be discussed more transparently in the manuscript. As it stands, the lack of this key active site cofactor could be missed because it is only briefly mentioned twice in the Results section. And, more importantly, many of the figures show the H-cluster as though it is present in the cryo-EM model. If this component of the active site is modeled based on its presumed location – rather than its observation in the cryo-EM map – then this distinction should be indicated as such in the figures. And the limitations of interpreting the structure due to the missing H-cluster should be more explicitly discussed. In the current draft, this important detail could be easily missed by the reader. It would be ideal to either obtain a structure of the protein reconstituted with the H-cluster or to further characterize the apo preparation to show that its cofactor content, properties, etc are not different from the active version containing the H-cluster.

We’d like to thank Reviewer 1 for their supportive and constructive review of our manuscript. Together with Reviewer 2, their suggestions have helped us to substantially improve the clarity and presentation of our work and we hope they find our revised manuscript to be acceptable for publication.

In our previous publication (Chongdar *et al.*, J. Biol. Inorg. Chem., 2020), we characterized both the apo-enzyme lacking [2Fe]_H_ and holo-enzyme after reconstitution of [2Fe]_H_ and showed that they contained the expected number of iron-sulphur clusters and had the expected spectroscopic properties based on what was known about the native enzyme from *Thermotoga maritima*. For cryoEM, we used the same preparation.

We realise that this was not made clear enough in the original version of the manuscript and so we have made the following additions:

Page 3: “In our previous work, it was shown that this preparation contains all the redox cofactors of the native HydABC enzyme except for the [2Fe] subcluster of the hydrogenase active site (H-cluster), which *E. coli* is unable to synthesize. In particular, Fe quantitation measurements of the heterologously produced enzyme agreed with the expected number of iron-sulfur clusters based on sequence analysis, and were even higher than those from the native enzyme (Verhagen BBA 1999). Furthermore, EPR spectra of the reduced apo- and reduced holo-HydABC (where the H-cluster is EPR-silent) were identical to each other and the same as those from the native enzyme (Verhagen BBA 1999). A drawback of using this apo-HydABC preparation is that we cannot observe how the structure is affected by reduction by H_2_.”

We also added the following sentence on Page 14 of the Results: “Importantly, the structure around the H-cluster is highly conserved between CpI and HydABC with only very small deviations in the positions of serveral conserved side-chains (Figure 3—figure supplement 1).” and added a supplementary figure (Figure 3—figure supplement 1).

We have also amended the Figures so that instead of indicating that the H-cluster is present it is only [4Fe-4S]_H_ that is present and we have further explained the lack of the [2Fe]_H_ subcluster in the figure captions.

We have also explained in more detail how our structure allows us to exclude the H-cluster as the site of electron-bifurcation on Page 20: “Lastly, the H-cluster is located at the end of an electron transfer pathway rather in the middle of one, which makes it a very unlikely branch site.”

The premise of the paper relies heavily on the electron bifurcation chemistry proposed to be important in this enzyme. But the experimental basis for this phenomenon in this hydrogenase homolog is not clearly explained. It is stated on page 4, line 71, that the mechanism of HydABC is likely a flavin-based electron bifurcation one – but the experimental results that support this conclusion are not explained. It is important to understand the basis for this phenomenon in this enzyme given that the entire manuscript is centered on this idea.

We have now extended the discussion on how our structure agrees with the idea of the FMN serving a dual role as both an electron bifurcation center and a site of NADH oxidation/NAD^+^ reduction on Page 20: “By excluding that the H-cluster or a second flavin function as bifurcation sites, and since our new cryoEM structures reveal that there are no other possible electron bifurcation sites, we are left with the possibility that the FMN in HydB is indeed the electron-bifurcation site. Our first structure reveals that the FMN is located at a branch point connecting the core electron transfer pathway from the H-cluster and the additional iron-sulfur clusters B1 and C1, while our additional structures reveal that the FMN is close to a zinc site and a mobile iron-sulfur cluster domain, all indicating that it is ideally located for behaving as an electron bifurcation center. However, the FMN must bifurcate electrons in an entirely unprecedented way, since it must also serve as the two electron donor/acceptor of NAD^+^/NADH.”

Earlier this year Adams et al. published a series of cryo-EM structures of a structurally related NiFe hydrogenase that contains an HydABC module that is similar to the enzyme core reported here. This manuscript is cited twice and discussed briefly – mostly to highlight minor structural/cofactor differences. However, the Adams study seems to reach a similar conclusion – that the sole flavin cofactor is the site of bifurcation and that conformational changes modulate the properties of the FMN and other cofactors linked to it in the pathway. The common features of the two systems would seem to warrant a more extended discussion of the similarities.

We agree that a more detailed comparison of the features of the two systems would benefit the readership. As such we’ve added the following sentences on Page 21: “A similar mechanistic proposal was made by Feng et al. (Feng Sci Adv 2022) to explain electron bifurcation in the related [NiFe] hydrogenase (HydABCSL) from A. mobile. HydA, B and C in A. mobile are homologous to HydA, B and C in T. maritima, respectively, however, HydA in A. mobile lacks the H-cluster and instead the enzyme contains HydS and L, which form the [NiFe] hydrogenase unit. The fact that both enzymes bifurcate electrons, yet do not both contain the H-cluster, further supports the idea that the H-cluster is not the site of electron bifurcation in TmHydABC. Otherwise, the structures of the HydABC units in both enzymes are very similar. However, it was proposed that instead of a zinc site AmHydB contains an additional [2Fe-2S] cluster, which allows electron transfer between the site of ferredoxin oxidation in the B3/B4 clusters and the [2Fe-2S] cluster in AmHydC. The latter was also suggested to be located in a mobile domain and that conformational changes are triggered by events at the FMN site. However, the authors did not consider in detail how nucleotide binding or changes in the FMN redox potentials could be coupled to conformational changes. While the two mechanistic proposals differ in the details, they both consider the FMN and unique arrangement of metallocofactors around it to be crucial components for electron bifurcation.”

Throughout the manuscript, the authors use unconventional terminology that is difficult to understand. In the abstract, on line 27, the two heterotrimers are described as "electrically connected." I don't understand what this term means – but the authors use it repeatedly in the main body of the manuscript. In some cases, I would guess that the intended meaning refers to two cofactors that are close enough for single step protein-mediated electron transfer (within >15-20 Å) – but it is difficult to tell because of the imprecision of the language.

We apologise for this terminology and we have removed “electrically” in all instances and tried to amended these sentences to be more precise. Essentially, by “electrical connection” we wanted to convey that electrons could be efficiently transferred between two parts of the electron transfer pathway, but we now realise that this is not precise terminology. For example, on Page 12 we changed “An electrical connection…” to “An electronic connection…” and on Page 14 we changed “…that electrically connects the [4Fe-4S] subcluster of the H-cluster (analogous to the cluster N7 in Tt complex I) with the rest of the electron transfer network.” to “…that connects the [4Fe-4S] subcluster of the H-cluster (analogous to the cluster N7 in Tt complex I) with the rest of the electron transfer network (<10 Å separation from both).”

In the discussion, on page 12 lines 309-311, the authors refer to electron transfer between "redox couples." This description also lacks precision. The "couple" does not transfer electrons. Electron transfer occurs between the cofactors. This section should be rewritten accordingly.

We apologise if the description appears to lack precision. We have rewritten the section as follows and hope that the reviewer finds this more appropriate: “The FMN in HydABC exchanges electrons with NAD^+^/NADH, which forms the high potential couple (Eº’ ≈ -320 mV), and exchanges electrons with the H-cluster, which in turn exchanges electrons with 2H^+^/H_2_, the intermediate potential couple (Eº’ ≈-420 mV), while oxidized/reduced ferredoxin, the low potential couple (Eº’ ≈ -450 mV), appears to exchange electrons with a separate pathway.”

Figure 5 is challenging to understand in its current form. The gray shading of certain boxes is not described in the figure or the caption. The text is very small. And the term "electron confurcation" is not defined well in the manuscript text.

We have amended Figure 5 to make it easier to understand and have added a sentence explaining what is meant by electron confurcation to both the introduction on Page 1, the discussion on Page 21, and in the figure legend to Figure 5. It was necessary to remove the mechanism during electron confurcation from the Figure in order to make the electron bifurcation mechanism larger but we have moved the electron confurcation part of the figure to Figure 5—figure supplement 1.

Reviewer #2 (Recommendations for the authors):The main concern is that many of the mechanistic discussions could potentially be strengthened by exploring further the enzyme structure (and its conformational changes) in the presence of its cofactors, particularly NAD+/NADH and ferredoxin. I fully appreciate that this work may be complicated by the binding of these cofactors being labile and being driven by specific redox state(s) of the enzyme clusters. I also appreciate that this would require a large amount of additional work. For this reason, I think that it is not appropriate to request that additional experiments are performed now, but it would be ideal if the authors expand the discussion to comment further. For example, would these experiments be feasible with cryo-EM? Have they been attempted without success? Are these planned for future work? If not, what alternative techniques can be used to prove/disprove the proposed mechanism?

We thank the reviewer for their supportive evaluation of our manuscript and we can assure them that their suggested experiments are indeed possible and under way. We plan to use cryoEM to investigate the holo-enzyme (containing [2Fe]_H_), the enzyme under hydrogen, the enzyme in the presence of ferredoxin, and in the presence of nucleotides. All of these we anticipate (and have preliminary data) will show interesting additional conformational changes that will help us further refine our mechanistic proposal. We intend to support these results with additional functional and spectroscopic experiments.

We initially refrained from adding too much of our future plans, but now we have included the following sentences on Page 22 to mention a few of our proposed targets and why we think they will be useful: “Further structural studies are also underway with holo-HydABC to investigate the precise structural details of the H-cluster, the effects of reduction by H_2_, as well as the conformational changes induced by nucleotide and ferredoxin binding. These findings will then be correlated with spectroscopic and functional information to provide a detailed understanding of the mechanism of electron bifurcation in this interesting enzyme.”

The absence of the [FeFe] subcluster of the H-cluster should be discussed more clearly. The authors should discuss in the manuscript if this was omitted because the sample is exposed to oxygen during cryo-EM processing? Is this unavoidable? If this decision was based on other factors, these should also be discussed in the manuscript.

This was also an important criticism of Reviewer 1, and we apologise for not stating more clearly why the enzyme lacking [2Fe]_H_ was used. Indeed, as the reviewer suggests, oxygen sensitivity was our primary motivation. However, we also anticipated that, just as has been observed in other [FeFe] hydrogenases (see Esselborn et al., Chem. Sci. 2016) the H-cluster environment is not affected by the presence of the [2Fe]_H_ site. Thus, we were satisfied that our structure was likely to represent the same structure as the holo-enzyme containing the [2Fe]_H_ subcluster. We are actively working on solving additional structures including that of the holo-enzyme but this requires cryoEM grid preparation to be carried out in an anaerobic glovebox and currently we do not have access to this, but will do very shortly.

As per the reviewer’s suggestion we modified a sentence on Page 3 of the introduction: “Here, we have used this heterologously expressed apo-HydABC to prepare the cryo-EM grids under air, as apo-HydABC lacking the [2Fe] subcluster is much less oxygen sensitive.”